# SequenceMatch: Imitation Learning for Autoregressive Sequence Modelling with Backtracking

**Chris Cundy**[1]  **Stefano Ermon**[1]

[1]Department of Computer Science, Stanford University
{cundy, ermon}@cs.stanford.edu

## Abstract

In many domains, autoregressive models can attain high likelihood on the task of predicting the next observation. However, this maximum-likelihood (MLE) objective does not necessarily match a downstream use-case of autoregressively generating high-quality sequences. The MLE objective weights sequences proportionally to their frequency under the data distribution, with no guidance for the model's behaviour out of distribution (OOD): leading to compounding error during autoregressive generation. In order to address this compounding error problem, we formulate sequence generation as an imitation learning (IL) problem. This allows us to minimize a variety of divergences between the distribution of sequences generated by an autoregressive model and sequences from a dataset, including divergences with weight on OOD generated sequences. The IL framework also allows us to incorporate backtracking by introducing a `backspace` action into the generation process. This further mitigates the compounding error problem by allowing the model to revert a sampled token if it takes the sequence OOD. Our resulting method, SequenceMatch, can be implemented without adversarial training or architectural changes. We identify the SequenceMatch-$\chi^2$ divergence as a more suitable training objective for autoregressive models which are used for generation. We show that empirically, SequenceMatch training leads to improvements over MLE on text generation with language models and arithmetic.

## 1 Introduction

Autoregressive models such as the GPT series of causally masked transformers (Brown et al., 2020; Radford et al., 2018) are able to perform a variety of downstream tasks such as question-answering, translation, and summarization, after simply training on a large corpus of text with the objective of predicting the next token given the previous tokens. However, autoregressive language models suffer from a variety of pathological behavior when deployed on the task of free-form text generation (Holtzman et al., 2019; Welleck et al., 2019), particularly at lower generation temperatures or with smaller models. These include generating the same token or series of token repeatedly, or generating gibberish outputs. This problem of neural text degeneration has been linked to the training objective for LLMs, which trains a conditional distribution for the next token given a (partial) sentence (Fu et al., 2021). When deployed in an autoregressive fashion, the model has its own outputs as inputs, resulting in a compounding error problem that rapidly takes the model out of distribution (OOD). This compounding error problem is also a key issue in the imitation learning subfield of reinforcement learning, where the goal is to learn a policy (a distribution over next actions given the past) which results in trajectories similar to a set of expert trajectories. The approach of directly matching the expert's actions leads to a compounding error (Ross et al., 2011), which has led to several works proposing to address this problem by minimizing alternative divergences (Arora et al., 2022; Shi et al., 2018). These alternative divergences encourage the policy to return to expert states if the generated trajectory starts to diverge from them. We argue that two key issues can prevent autoregressive models trained with maximum-likelihood from generating fluent sequences at evaluation time. First is the divergence measure used to evaluate the difference between the model and the data distribution. Because the MLE loss does not have any contribution from OOD sequences, the behavior of

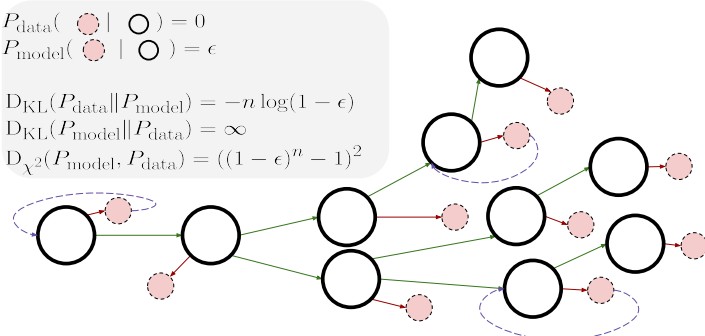

Figure 1: A toy model of an autoregressive generation problem, such as language modelling. Our task is to learn a set of conditional distributions that continue the sequence similarly to those sequences in the dataset (green arrows), and avoid incorrect next tokens (red arrows). Our method trains against divergences that more heavily punish out-of-distribution sequences. We additionally introduce a backspace action which can backtrack from an erroneous token (dashed purple arrows).

the model on OOD sequences is unconstrained. We address this by minimizing the $\chi^2$-divergence between a mixture of the data and autoregressively generated sequences. This divergence is known to perform much better than MLE in imitation learning (Garg et al., 2021; Al-Hafez et al., 2023).

Secondly, if a model generates an OOD token, there may be no natural continuation which is similar to sequences from the data distribution, and so the model may be unable to return to the data distribution even if our $\chi^2$-divergence encourages this. To address this, we augment the generation process with a backspace action `<bkspc>`, which deletes the previous token, and allows the model to correct for erroneous generations. By incorporating recent work in non-adversarial imitation learning (Garg et al., 2021), our method, *SequenceMatch*, is able to train autoregressive models against alternative divergences such as the $\chi^2$-mixture divergence while augmenting the policy with a backspace action. The SequenceMatch loss is a fully supervised loss without adversarial training, and can be applied on top of pretrained models as a finetuning step. To summarize our contributions:

- We formulate the sequence generation problem as an imitation learning (IL) problem, and formulate a general non-adverarial objective for minimizing divergences between occupancy measures based on (Garg et al., 2021), handling (among others) the forward and reverse KL-divergence, JS-divergence, and $\chi^2$ divergence.

- We develop a novel masking scheme allowing training of a transformer-based autoregressive model with a backspace action with no additional overhead vs MLE.

- Finally we evaluate the empirical performance of SequenceMatch-trained models, showing improved performance over the maximum likelihood objective in text generation and arithmetic.

## 2 PRELIMINARIES: TRAINING OBJECTIVES FOR AUTOREGRESSIVE MODELS

### 2.1 KL-DIVERGENCE

Typically, autoregressive models are trained against a maximum-likelihood objective. This objective can be motivated by treating our dataset as consisting of sequences of random variables $(x_1, \ldots, x_N)$, with a corresponding probability distribution $P_{\text{data}}(x_1, \ldots, x_N)$, with a fixed length $N$. The goal is to learn a parameterized model $P_\theta(x_1, \ldots, x_N)$ that is close to $P_{\text{data}}$. The KL-divergence between the data distribution and the model has a useful decomposition: $D_{\text{KL}}(P_{\text{data}} \| P_\theta) = -\mathbb{E}_{x_{1:N} \sim P_{\text{data}}} \left[ \sum_i^N \log P_\theta(x_i | x_{<i}) \right] + C$, where $C$ is a constant that does not depend on $\theta$. For a dataset $\mathcal{D} = \{x_{1:N}^j\}_{j=0}^{N_{\text{data}}}$ of sequences drawn i.i.d. from $P_{\text{data}}$, this can be approximated with an estimator $\hat{D_{\text{KL}}}(P_{\text{data}} \| P_\theta) = \frac{1}{N_{\text{data}}} \sum_j \sum_i^N \log P_\theta(x_i^j | x_{<i}^j) + C'$. Hence, minimizing the KL-divergence is equivalent to maximizing the model's (log-) likelihood of the next element in

the sequence given the previous elements. The (typically unknown) density under the data distribution $P_{\text{data}}$ is not required. In some domains, the length of the sequences $n_j$ differs in each example $j$, which can be incorporated by choosing an effective length $N = \max n_i$, and treating all sequences shorter than $N$ as having a sequence of padding tokens appended[1]. In the sequel with some abuse of notation we will write $P_{\text{data}}(s_n)$ for $\sum_i^n \log P_\theta(x_i|x_{<i})$, for partial sequences that may not terminate until after $n$, and $P_{\text{data}}(x|s_n)$ to signify the probability of the next token given a partial sequence.

### 2.1.1 LIMITATIONS OF THE KL-DIVERGENCE

However, while it is clear that minimizing the KL-divergence will result in $P_\theta = P_{\text{data}}$ (for a sufficiently flexible parameterization $P_\theta$), it is not obvious what the behaviour is of models which *approximately* minimize the KL-divergence. In figure 1, a chain distribution is shown, with sequences of length $N$. The model $P_\theta$ has an $\epsilon$ error on each conditional, where an error leads to an OOD sequence which has no support under the data distribution. This leads to an error in the KL metric of order $n\epsilon$. However, the probability of getting to the end of the chain before an incorrect token is picked is $1 - (1 - \epsilon)^n$, and so the value of the KL-divergence is not a good metric if our main quantity of interest is how often generated sequences are in-distribution. Furthermore, the KL-divergence weights the loss by the frequency under the data distribution, leaving the model's behavior out-of-distribution from the data essentially undetermined.

In non-autoregressive generative modelling, several different divergences are commonly used, such as the Wasserstein distance (Arjovsky et al., 2017) and Fisher divergence (Song & Ermon, 2019). Particularly interesting is the $\chi^2$ divergence $D_{\chi^2}(P_\theta, P_{\text{data}}) = \mathbb{E}_{x \sim P_{\text{data}}}\left[(P_\theta(x)/P_{\text{data}}(x) - 1)^2\right]$. Indeed, figure 1 shows that the $\chi^2$-divergence in this case is equal to the squared probability of staying in the data distribution of sequences. We can further penalize out-of-distribution behavior by considering the divergence between mixtures $D_{\chi^2}(P_\theta, (P_{\text{data}} + P_\theta)/2)$, as we do in our practical algorithm. However, it is difficult in practice to compute any divergence involving the density of the data, which must be substituted for with an approximation from a discriminator.

In reinforcement learning, there are several methods which can minimize diverse divergences between the distribution of trajectories from an expert and a learned policy. The approaches are non-adversarial, even with unknown expert density (Garg et al., 2021; Swamy et al., 2021; Barde et al., 2020; Al-Hafez et al., 2023). A key feature of these methods is that they operate on *occupancy measures* instead of joint distributions, a concept which we describe in the next section.

## 3 METHOD

### 3.1 SEQUENCE MODELLING AS A MARKOV DECISION PROCESS

We consider a sequence modelling problem represented as a Markov decision process (MDP), defined by a tuple $(\mathcal{S}, \mathcal{A}, \mathcal{P}, r, \gamma)$, with state and action spaces $\mathcal{S}, \mathcal{A}$, dynamics $\mathcal{P}(s'|s, a)$, reward function $r(s, a)$, and discount factor $\gamma \in (0, 1)$. In our case, the state space $\mathcal{S}$ is the set of all sequences (of all lengths) with elements in a finite set $X$, the vocabulary. The action set $\mathcal{A}$ is a finite set. For concreteness, we can assume that $X \subseteq \mathcal{A}$ (i.e. we have an `insert-token` action for each token), as well as an additional backspace action `<bkspc>`. In our case, the initial state is given by a special `<bos>` token, while the dynamics for an `insert-token` action in a state (sequence) $s$ leads deterministically to the sequence $s'$ consisting of $s$ with the given token appended.

Combined with a policy $p_\theta(a|s)$, the MDP defines a distribution over (possibly infinite-length) sequences, following the generative process of sampling an action $a \sim p_\theta(\cdot|s)$, then sampling the next state $s' \sim \mathcal{P}(s'|s, a)$, etc. Finally, we assume that a special end of sequence token `<eos>` induces a terminal state: in a state $s$ with `<eos>` as the final element, all actions cause a self-transition to $s$. This probabilistic process reduces exactly to the autoregressive formulation of sequence modelling when the action set is the same as the vocabulary. A backspace action in a state $s$ deterministically transitions to a state $s'$ with the final token in the sequence $s$ removed.[2] An example of the states and actions can be seen in figure 2. The MDP framework formalizes the intuitive picture in figure 1: language modelling can be viewed as the traversal of a tree where the nodes are (partial) sequences.

---

[1]Some care is required, as averaging the loss of each example over its length gives an inconsistent estimator.
[2]In the case of $s = $ `<bos>` and `<bkspc>` is used, $s' = $ `<bos>` also

A central quantity of interest is the occupancy measure. We denote by $s_t$ the random variable consisting of the state at time $t$ under a policy $p(a|s)$ and the MDP defined above. Then, the occupancy measure $\rho(s, a) : \mathcal{S} \times \mathcal{A} \to [0, 1]$ is the (discounted) probability of observing a particular sentence $s$ at time $t$ and taking action $a$ given that sentence:

$$\rho(s, a) = (1 - \gamma)p(a|s) \sum_t \gamma^t P(s_t = s). \tag{1}$$

In other words, the occupancy measure is proportional to the observed frequency of a particular (sentence, next-action) pair occurring, with occurrances discounted in time by a factor of $\gamma$. In the absence of editing actions, $\mathcal{A} = X$ and the occupancy measure is a discounted probability over (partial) sequences: for a sequence $s_n$ of length $n$, $\rho_{\text{data}}(s_n, x) = (1 - \gamma)\gamma^n P_{\text{data}}(s')$, where $s'$ is the sequence obtained by appending $x$ to $s$. Given editing actions which can reduce the length of a sequence, the occupancy measure becomes more complicated, as the same sequence can occur at multiple time indices. For a function $r$, the expectation with respect to $\rho$ has the usual meaning: $\mathbb{E}_{(s,a)\sim\rho}[r(s,a)] = \sum_{\mathcal{S},\mathcal{A}} \rho(s,a)r(s,a)$, with sum over the discrete action space and (countably infinite) state space. Occupancy measures provide an alternative way of modelling sequences, allowing us to impose a measure over all sequences, even in the presence of editing actions. The next section illustrates that we can non-adversarially minimize a large variety of divergences between occupancy measures, compared to only the KL divergence in the typical joint probability formulation.

## 3.2 MINIMIZING OCCUPANCY DIVERGENCES

Our aim is to learn a policy $p_\theta(a|s)$ which induces an occupancy measure $p_\theta$ such that it is close to the data occupancy measure $p_{\text{data}}$. We define the data occupancy measure by forming the data policy $p_{\text{data}}(a|s)$ corresponding to the conditionals $P_{\text{data}}(x|s_n)$ and setting the probability of editing actions to zero. It is known that matching occupancy measures implies matching policies: if $\rho_\theta = \rho_{\text{data}}$ for a valid occupancy $\rho_\theta$, then the corresponding $p_\theta(a|s) = P_{\text{data}}(a|s)$ (Syed et al., 2008). Therefore, it is reasonable to minimize divergences between occupancy measures. We extend the derivations in Garg et al. (2021) to the case with infinite-dimensional state space. We consider distances between occupancy divergences parameterized by the following form:

$$d_\psi(\rho_\theta, \rho_{\text{data}}) = \sup_{r \in \mathcal{R}} \mathbb{E}_{(s,a)\sim\rho_\theta}[r(s,a)] - \mathbb{E}_{(s,a)\sim\rho_{\text{data}}}[r(s,a)] - \psi(r), \tag{2}$$

where $\psi$ is a convex regularizer. The critic $r$ picks out differences between the occupancies, while if $\rho_\theta = \rho_{\text{data}}$, the difference in expectations will be zero for any $r$. This family of divergences includes all $f$-divergences such as the KL and JS-divergence, as well as the Wasserstein distance and MMD, as described in the appendix. The problem can be made tractable by adding an entropy term:

$$\inf_\theta d_\psi(\rho_\theta, \rho_{\text{data}}) - \alpha H[\rho_\theta], \tag{3}$$

with the entropy $H[\rho_\theta] = -\mathbb{E}_{(s,a)\sim\log\rho_\theta}[\log \rho_\theta(s,a)]$, and $\alpha$ a chosen regularization strength. Substituting in the definition of $d_\psi$, we obtain the min-max problem $\inf_{\rho_\theta} \sup_r L(\theta, r) = \inf_{\rho_\theta} \sup_r r \cdot (\rho_\theta - \rho_{\text{data}}) - \psi(r) - \alpha H[\rho_\theta]$. We prove in the appendix that the saddle-point property in Ho & Ermon (2016) extends to our infinite-dimensional case, so $\inf_{\rho_\theta} \sup_r L(\theta, r) = \sup_r \inf_{\rho_\theta} L(\theta, r)$. We can interpret the outer maximization as finding a critic (LeCun et al., 2006) $r$ for sequences and actions $s, a$ such that the model has high values on examples from the dataset and low values on the examples from the learned model. The inner minimization over $\theta$ is an entropy-regularized minimization of the KL-divergence between $\rho_\theta$ and $r$. Approaching this directly by explicitly learning $r$ and $\rho_\theta$, would lead to an objective similar to a GAN (Goodfellow et al., 2014). This is known to be difficult to train (Jabbar et al., 2020). Instead, we can solve the problem with optimization over a single variable by a transformation of variables. In the following section, we recover an objective $\mathcal{J}$ which is equivalent to the objective in equation 3, but only involves optimization over the logits of a policy. For ease of exposition, $\alpha = 1$ in the next section.

### 3.2.1 REFORMULATING THE OCCUPANCY DIVERGENCE MINIMIZATION PROBLEM

We first introduce the $Q$-function, corresponding to the discounted rewards obtained in state $s$ by taking action $a$. Formally, it is the unique fixed point of the soft Bellman operator $\mathcal{B}_r^\theta$, where $\mathcal{B}_r^\theta Q(s,a) = r(s,a) + \gamma \mathbb{E}_{s'\sim\mathcal{P}(s,a)}[V^\theta(s')]$, for the value function $V^\theta(s) =$

$\mathbb{E}_{a \sim p_\theta(\cdot|s)}[Q(s,a) - \log p_\theta(a|s)]$. The inverse Bellman operator $\mathcal{T}^\theta$ is the inverse of this operator, given by $(\mathcal{T}^\theta Q)(s,a) = Q(s,a) - \gamma \mathbb{E}_{s' \sim \mathcal{P}(s,a)}[V^\theta(s')]$. For a fixed policy $\theta$, there is a one-to-one correspondence between $r$ and $Q$ via the Bellman and inverse Bellman operators (proved in the appendix). Crucially, for the unique occupancy $\rho^*$ which solves $\max_\theta \mathbb{E}_{s,a \sim \rho_\theta}[r(s,a)] - H[\rho_\theta]$, the optimal policy $\log p^*(a|s)$ corresponding to $\rho^*$ is proportional to the corresponding $Q$-values $Q^*$: $\log p^*(a|s) = Q^*(s,a) - V^{\theta^*}(s) = Q^*(s,a) - \log \sum_{a' \in \mathcal{A}} \exp Q^*(s,a')$. The key idea of the following derivations is that the optimal policy is uniquely determined by the optimal $Q$-values, while the reward is determined by the $Q$-values. This allows us to optimize solely over $Q$-values.

**Proposition 3.1.** *The following equalities hold for the loss:*

$$
\begin{aligned}
\inf_\theta d_\psi(\rho_\theta, \rho_{data}) - H[\rho_\theta] &= \sup_r \inf_\theta \mathbb{E}_{\rho_{data}}[r(s,a)] - \mathbb{E}_{\rho_\theta}[r(s,a)] - H[\rho_\theta] - \psi(r), \\
&= \sup_Q \inf_\theta \mathbb{E}_{\rho_{data}}[\mathcal{T}^\theta Q] - \mathbb{E}_{\rho_\theta}[(\mathcal{T}^\theta Q)] - H[\rho_\theta] - \psi(\mathcal{T}^\theta Q), \\
&= \sup_Q \inf_\theta \mathbb{E}_{\rho_{data}}[\mathcal{T}^\theta Q] - (1-\gamma)\mathbb{E}_{\mathcal{P}_0}[V^\theta(s_0)] - \psi(\mathcal{T}^\theta Q), \\
&= \sup_Q \inf_\theta \mathbb{E}_{\rho_{data}}[\phi(Q(s,a) - \gamma\mathbb{E}_\mathcal{P}[V^\theta(s')])] - (1-\gamma)\mathbb{E}_{\mathcal{P}_0}[V^\theta(s_0)], \\
&= \sup_Q \inf_\theta \mathbb{E}_{\rho_{data}}[\phi(Q(s,a) - \gamma\mathbb{E}_\mathcal{P}[V^\theta(s')])] - \mathbb{E}_\rho[V^\theta(s) - \gamma V^\theta(s')], \\
&= \sup_Q \mathbb{E}_{\rho_{data}}[\phi(Q(s,a) - \gamma\mathbb{E}_\mathcal{P}[V(s')])] - \mathbb{E}_\rho[V(s) - \gamma V(s')],
\end{aligned}
$$

*where $\phi$ is concave, $\mathbb{E}_{\rho_{data}}$ denotes expectations over sampled states and actions $s, a$, $\mathbb{E}_\mathcal{P}$ denotes an expectation over successor states $s'$, and $\mathbb{E}_\rho$ denotes an expectation over sampled states $s$ and successor states $s'$, for any occupancy $\rho$. $V(s)$ (without $\theta$) is given by $V(s) = \log \sum_{a' \in \mathcal{A}} \exp Q(s,a')$.*

*Proof.* The full proof is given in the appendix. As a sketch, the first equality holds from the previous section. The second is obtained by replacing $r$ with $\mathcal{T}^\theta Q$ and verifying that the two optimization problems are equal. The third line is via a telescoping sum argument first described in (Kostrikov et al., 2019). In the fourth line we replace $\psi(r)$ with a simpler regularizer $\mathbb{E}_{s,a \sim \rho_{data}}[g(r(s,a))]$, where $g(r) = r - \phi(r)$ if $r \in \Omega$, and infinity otherwise. The fifth line follows from expanding the telescoping sum in a different way, incorporating samples from any policy. In the final line we parameterize the policy from the $Q$-values, setting $\log p_Q(a|s) = Q(s,a) - \log \sum_{a' \in \mathcal{A}} \exp Q(s,a')$. We then show that the optimization problem over $(Q, p_Q)$ has the same optimum as the optimization over $(Q, \theta)$, so we can optimize solely over $Q$. $\qquad\square$

We relabel $Q$ as $\ell_\theta$ to make the connection to logits clearer, resulting in the fully supervised objective over logits $\ell_\theta$: $\mathcal{J}(\ell_\theta) = \frac{1}{\alpha}\mathbb{E}_{\rho_{data}}[\phi(\alpha\ell_\theta(a|s) - \alpha\gamma V(s')] - \frac{1}{2}\mathbb{E}_{\rho_{data}}[V(s) - \gamma V(s')] - \frac{1}{2}\mathbb{E}_{\rho_\theta}[V(s) - \gamma V(s')]$, where $(s,a,s') \sim \rho$ corresponds to sampling $s, a$ from $\rho$ and $s'$ from $\mathcal{P}(\cdot|s,a)$, and $V(s) = \log \sum_{a' \in \mathcal{A}} \exp \ell_\theta(a'|s)$. Minimizing this objective is equivalent to $\min_\theta d_\psi(\rho_\theta, \rho_{data}) - \alpha H[\rho_\theta]$, where $d_\psi(P,Q) = \sup_{r \in \Omega} \mathbb{E}_{x \sim P}[\phi(r(x))] - \mathbb{E}_{x \sim Q}[r(x)]$. By choosing $\Omega$ and $\phi$, we can recover $f$-divergences, including KL, JS and $\chi^2$ divergences, and additionally the Wasserstein and MMD distances. The corresponding choices are given in the appendix.

## 4 PRACTICAL OCCUPANCY MATCHING WITH AUTOREGRESSIVE MODELS

In practice, we wish to train a parameterized model $p_\theta(a|s)$ which can serve as a policy, emitting a probability distribution over the next action given a partially completed sequence $s$. A typical choice is a transformer (Vaswani et al., 2017): with parameters $\theta$ it gives a distribution over the next token $x_i$ given the previous tokens $x_{<i}$, parameterized as unnormalized logits $\ell_\theta$. Thus the MLE loss, with a complete sequence $x_{1:N}$, can be written as $\hat{\mathcal{L}}_{\text{MLE}}(\ell_\theta) = \sum_{i=1}^N \ell(x_i|x_{<i}) - \log \sum_{x' \in X} \exp \ell(x'|x_{<i})$, or $\hat{\mathcal{L}}_{\text{MLE}}(\ell_\theta) = \sum_{i=1}^N \ell(x_i|x_{<i}) - V(x_{<i})$.

To form an estimator for the loss derived in the previous section, samples from $\rho_\theta$ are required. We obtain these samples by sampling complete sequences from the policy autoregressively and weighting the partial sequence at time $t$ by a factor of $\gamma^t$. We similarly sample sequences from $\rho_{data}$

by sampling complete sequences from $P_{\text{data}}$ and weighting. So, for a length-N sequence of states $s_{1:N}$ from a dataset, corresponding actions $a_{1:N}$ and a generated length-$M$ sequence $u_{1:M}$ of states from the model, we can form an estimator for the loss from the previous section:

$$
\hat{\mathcal{J}}(\ell_\theta) = \underbrace{\sum_i^N \gamma^i \frac{1}{\alpha} \phi\left(\alpha\ell_\theta(a_i|s_i) - \gamma\alpha V(s_{i+1})\right)}_{\text{Penalized difference from action logit to next state value}} - \underbrace{\sum_i^N \frac{\gamma^i}{2}\left[V(s_i) - \gamma V(s_{i+1})\right]}_{\text{State, next state value difference under data}}
$$

$$
- \underbrace{\sum_i^M \frac{\gamma^i}{2}\left[V(u_i) - \gamma V(u_{i+1})\right]}_{\text{State, next state value difference under model}} + \underbrace{\frac{\gamma^N}{\alpha(1-\gamma)}\phi\left(\alpha(1-\gamma)V(s_N)\right) - \frac{\gamma^N}{2}V(s_N) - \frac{\gamma^M}{2}V(u_M)}_{\text{Loss from completed sequences}},
$$

(4)

and $V(s) = \log\sum_{a'\in\mathcal{A}}\exp\ell_\theta(a'|s)$. The separate treatment of the `<eos>` tokens arises from taking the sum over the infinite timesteps in equation 1 in the terminal states. As shown in the appendix, the estimator is unbiased and consistent for finite $\phi$. It has also been shown (Al-Hafez et al., 2023) that minimizing the mixture divergence $D_{\chi^2}(\rho_{\text{data}}, (\rho_{\text{data}} + \rho_\theta)/2)$ is more effective than simply minimizing the $\chi^2$-divergence between model and data. This can be implemented by calculating the loss for the $\chi^2$-divergence (with $\phi(x) = x - \frac{1}{4}x^2$) and adding an additional regularization term $\mathbb{E}_{s,a,s'\sim\rho_\theta}\left[(\alpha\ell_\theta(a|s) - \gamma\alpha V(s'))^2\right]$. We show in the appendix that with no backspace actions, $\lim_{\alpha\to 0}\mathcal{J}_{\ell_\theta} = D_{\text{KL}}(\rho_{\text{data}}\|\rho_\theta)$ reduces to a $\gamma$-reweighted MLE objective.

## 4.1 Efficient Training

Editing actions which can delete previous parts of the input are challenging to implement while retaining the fast training of transformer-based autoregressive models. For instance, the sequence of actions `[a; b; <bkspc>]` cannot be fed directly into a policy network $p_\theta(a|s)$, since it contains actions, not states. The sequence `[a; b; <bkspc>]` is not a valid state: the corresponding state is `[<bos> a]`. In order to convert this into a form where we can compute the relevant logits using masked attention, we must pre-process the sequence of actions into corresponding inputs, labels, masks and position IDs using algorithm A in the appendix. The preprocessing is illustrated in figure 2. On the other hand, generation with backspace actions is straightforward: we already keep previous key-value cached

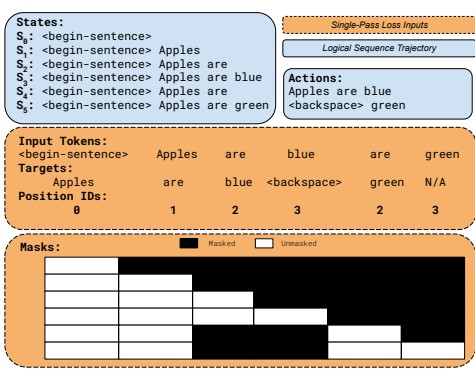

Figure 2: Transforming states and actions to single-pass inputs for parallel training.

values for generation with transformers. When `<bkspc>` is sampled, we simply roll back the state of the key-value cache and position id with negligible overhead. Additionally, the loss requires sampling from the model during training, which is typically slow. However, the sequences do not need to be exactly sampled from the current policy. Since any policy can be used, sequences generated from the policy at previous training steps are stored in a replay buffer (Mnih et al., 2013) and reused. We give an empirical analysis of the overhead when using SequenceMatch in the appendix.

## 4.2 Augmenting Expert Sequences with Backspace

To provide the policy with examples of how the `<bkspc>` action should be used, we augment the data sequences as follows: with (small) probability $\eta$, we replace a sequence $\ldots, x_{i-1}, x_i, x_{i+1}, \ldots$ with $x_{i-1}, x_i, x_i', $ `<bkspc>`$, x_{i+1}, \ldots$, where $x_i'$ is chosen randomly from the vocabulary. However, we keep the action at position $i$ as $x_{i+1}$, with the result that the overall MDP is augmented with a stochastic dynamics: with probability $\eta$ a random token is inserted, instead of the chosen action. This is also applied to sequences which exceed the context length: the action is kept the same but the next token is forced to be the `<eos>` token. This introduces bias, as the policy learns to match the data distribution under a slightly different MDP than generation takes place in. In practice however, it leads to improved performance compared to the policy learning with no examples of `<bkspc>`.

---

**Algorithm 1:** Training an autoregressive model against a SequenceMatch objective

---

**Input** : Dataset $\mathcal{D}$ of data sequences, gradient-based optimizer step, number of train steps $n_{\text{train}}$,
        parameters $\alpha, \beta, \gamma, \phi$, sampling interval $k_{\text{sample}}$, fixed context length $T$
Add noise and process data sequences with algorithm A to form new effective trajectories
Initialize buffer $\mathcal{B}$ of model sequences; Initialize autoregressive policy $\ell_\theta(\cdot|s)$
**for** $k$ *in* $n_{train}$ **do**
    **if** $k \mod k_{sample} = 0$ **then**
        Populate $\mathcal{B}$ with trajectories $\mathcal{T} \sim \ell_\theta$; Process added sequences with algorithm A
        Remove oldest model sequences from $\mathcal{B}$
    **end**
    Sample dataset trajectories $\mathcal{T}_{\text{data}} \sim \mathcal{D}$ and model trajectories $\mathcal{T}_{\text{model}} \sim \mathcal{B}$
    Compute $g = \nabla_\theta \hat{\mathcal{J}}(\ell_\theta, \alpha, \gamma, \mathcal{T}_{\text{data}}, \mathcal{T}_{\text{model}})$ and update $\theta$ via step using gradient $g$
**end**

---

## 5   RELATED WORK

**Text Degeneration in Large Language Models**

In natural language processing the phenomenon of *text degeneration* can occur, when a language model produces repetitive or nonsensical sequences (Holtzman et al., 2019). Many explanations have been proposed to explain this phenomenon (Fu et al., 2021; Welleck et al., 2019); a leading theory is that the large vocabulary size induces the model to over-estimate the probability of OOD tokens. Once these tokens are sampled, the model's context is now out-of-distribution. Measures to mitigate this problem include top-$k$ sampling (Fan et al., 2018), restricting generations to the $k$ most likely tokens, and top-$p$ sampling (Holtzman et al., 2019), an adaptive variant of top-$k$ sampling. In addition, alternative training measures have been proposed to reduce the probability of the model producing OOD tokens. Unlikelihood training (Welleck et al., 2019) is discussed in detail in the appendix, while contrastive methods have also been proposed (Jiang et al., 2022), which force the representations of repetitive text to be far from the representations of correct text.

**Matching Divergences in Imitation Learning**

In the imitation learning(Ng & Russell, 2000) subfield of RL, the objective is to learn a policy giving a distribution over actions in each state, such that the distribution over trajectories is close to distribution of provided expert trajectories. A simple approach is behavioral cloning (Esmaili et al., 1995), which maximizes the likelihood of the expert's chosen actions, on average over the states that the expert is in. However, it has been shown (Ross et al., 2011) that this approach results in a *compounding error* problem, where the further the trained model gets from the typical expert states, the worse the model performs, incurring increasing error. Ho & Ermon (2016) show that minimizing the occupancy divergence between the expert and a learned policy could be written as a two-variable saddle-point problem. This more sophisticated method can take the dynamics of the problem into account, learning a policy which can return to the typical expert states if it erroneously leaves them. In Garg et al. (2021), this was further developed via a change of variables to only require a non-adversarial optimization over one variable. We can view our approach as a specialization of the IQ-Learn algorithm in Garg et al. (2021) to autoregressive sequence models.

## 6   EXPERIMENTS

In this section we demonstrate that SequenceMatch can be used with large, state-of-the-art models, and that it can be useful for downstream applications. The experiments also allow some insight into the relative importance of the different components of SequenceMatch, namely the `<bkspc>` token and the alternative loss. The first experiment shows that SequenceMatch can improve accuracy on a downstream task, and that it can detect OOD states. The second experiment shows that Sequence-Match training can generate sequences with higher similarity to the data compared to a variety of baselines. In the appendix, section G, we describe three additional experiments, on translation, multiplication, and prime factorization. In all experiments, we finetune Llama2-7b (Touvron et al., 2023), using quantized low-rank adapters (Dettmers et al., 2023). In addition to the adapters, a row is added to the unembedding layer for the `<bkspc>` token, and the unembedding layer is trained.

## 6.1 ARITHMETIC

We first examine the potential for SequenceMatch to improve the performance on downstream tasks. The dataset is the arithmetic add-or-sub-in-base sub-task of the math-dataset (Saxton et al., 2018), consisting of questions such as `In base 7, what is -1240 - -4156?`. We compare a maximum-likelihood model, a behavioral cloning model, and a SequenceMatch model, with varying levels and types of noise. 'Random noise' is generated by sampling a token at random from the vocabulary, and hence is not very likely to be a common mistake made when solving the problem. 'Ground-truth noise' is generated by sampling a token from the set $\{0, \ldots, b-1\}$, where $b$ is the base in the question. The latter type of noise is expected to generate sequences that are far closer to the type of inaccuracies that a model (or human) might make while solving the problem. However, the random noise is generic to all tasks, while the ground-truth noise must be constructed for a given task manually. Both types of noise are only added to the solution digits, not the question digits. We use a small dataset of only 5,000 questions, to demonstrate the improved generalization abilities of SequenceMatch. The prompts for the SequenceMatch generations are taken from the training dataset and truncated at the end of the question. The accuracy is computed over a held-out test set of 200 questions, and error bars obtained by training two models with different random seed.

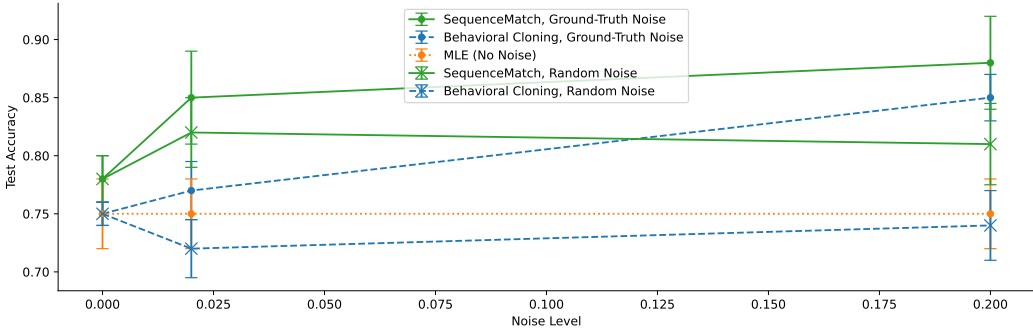

Figure 3: Accuracy on the arithmetic task against noise level (frequency with which noise tokens are added), for ground-truth noise consisting of digits and random noise consisting of random tokens. The ground-truth noise improves accuracy for the behavioral cloning and SequenceMatch models. The random noise does not improve performance for the behavioral cloning model, but does somewhat for the SequenceMatch model, likely helping the model to learn the dynamics of `<bkspc>`.

### 6.1.1 RESULTS

The results are shown in figure 3. It is clear that the models trained under the SequenceMatch loss outperform both the behavioral cloning (BC) and MLE baselines. With zero added noise, the improvement is small, as the model is not able to learn the backspace dynamics with no demonstrations from the expert. However, with a small amount of added noise, the SequenceMatch model has demonstrations of the use of `<bkspc>` and can start learning the dynamics through sampling. The BC model improves performance at higher levels of ground-truth noise, as the high level of noise acts as strong data augmentation. Indeed, this improved performance seems roughly proportional to the amount of added noise, as expected. Similarly, while random noise is useful to the SequenceMatch model due to the dynamics information, it does not help the BC model.

Qualitatively, the SequenceMatch-trained model is able to detect when it is out-of-distribution very accurately. In table 1 the best-performing SequenceMatch model is prompted with partial solutions which are incorrect (the first four in the test set with added random digit tokens). In each case it is able to detect the OOD case and return closely to the correct answer.

### 6.2 TEXT GENERATION

We use the same model and architecture as the previous section. Sequences are drawn from the open-webtext dataset[3], an open-sourced dataset similar to the training set for GPT-2 (Radford et al., 2018),

---

[3] https://github.com/jcpeterson/openwebtext

| Ground Truth QA | Prompt with Mistake | Completion Actions | Final State |
|---|---|---|---|
| In base 2, what is -11101111011001100 + 10100? Solution: -11101111010111000 | In base 2, what is -11101111011001100 + 10100? Solution: -1011 | `<bkspc><bkspc><bkspc>` `1101111<bkspc>1011000010` `<eos>` | -11101111011000010 |
| In base 8, what is 4354 + 33? Solution: 4407 | In base 8, what is 4354 + 33? Solution: 5 | `<bkspc>4417<eos>` | 4417 |
| In base 8, what is -4 + -576122? Solution: -576126 | In base 8, what is -4 + -576122? Solution: -374 | `<bkspc><bkspc><bkspc>` `576126<eos>` | -576126 |
| In base 5, what is 10 - 3311121? Solution: -3311111 | In base 5, what is 10 - 3311121? Solution: -31 | `<bkspc>31<bkspc>11111<eos>` | -3311111 |

Table 1: Mistake-conditioned completions for the arithmetic task. We add a random set of digit tokens to the prompt and generate from the SequenceMatch-trained model. The SequenceMatch model correctly deletes the initial tokens in all cases and eventually generates the correct solution in three of four cases.

| Model | MLE | MLE + C.S | MLE + ULK | MLE + `<bkspc>` | SequenceMatch |
|---|---|---|---|---|---|
| MAUVE ($\uparrow$) | $0.85 \pm 0.03$ | $0.86 \pm 0.03$ | $0.89 \pm 0.02$ | $0.84 \pm 0.02$ | $\mathbf{0.91 \pm 0.02}$ |
| n-gram $\mathbb{H}$ ($\uparrow$) | $4.57 \pm 0.02$ | $4.43 \pm 0.02$ | $4.57 \pm 0.01$ | $4.59 \pm 0.01$ | $\mathbf{4.60 \pm 0.01}$ |
| Diversity ($\uparrow$) | $0.56 \pm 0.02$ | $0.35 \pm 0.03$ | $\mathbf{0.57 \pm 0.01}$ | $0.56 \pm 0.01$ | $0.56 \pm 0.01$ |
| Perplexity ($\downarrow$) | $\mathbf{6.99 \pm 0.02}$ | N/A | $7.10 \pm 0.02$ | $7.02 \pm 0.02$ | $7.13 \pm 0.03$ |

Table 2: A Llama-2-7b model fine-tuned on the openwebtext dataset with different training and sampling regimes. Error bars are over two draws of 1000 evaluation samples. C.S. and ULK are contrastive sampling and unlikelihood loss training, respectively. The SequenceMatch model achieves the highest MAUVE score and n-gram entropy, with diversity very close to the best value from the MLE + unlikelihood training.

with a 1024-length context window, truncating sequences that are longer. The model-generated trajectories are generated from examples from the training dataset, with a prompt length randomly chosen with a maximum of 256. The generated sequences have a max length of 512 (although they may terminate earlier). We compare a SequenceMatch-trained model against several baselines: a model trained against the typical MLE objective, and a behavioral cloning model trained with injected noise and `<bkspc>` labels. We also test `MLE + C.S.`, which is MLE with contrastive sampling (Jiang et al., 2022). Finally, `MLE + ULK` is maximum-likelihood with the unlikelihood token loss (Welleck et al., 2019). We train for 5,000 gradient steps. The SequenceMatch parameters are set to $\alpha = 0.01$, $\eta = 0.001$ and $\gamma = 0.998$. Our main metric for quality of generations is the MAUVE score (Pillutla et al., 2022), a non-parametric method for evaluating the quality of a generative model. The MAUVE score is formed by taking a low-dimensional PCA of an embedding of the generated sequences. The score is a mixture of forward and reverse KLs between data and model-generated sequences, between zero and one (higher is better). Additionally we report the n-gram entropy and the diversity metric (Jiang et al., 2022), given by $\prod_{n=2}^{4} \left(1.0 - \frac{\text{rep-n}}{100}\right)$, where rep-n $= 100 \times \left[1.0 - \frac{|\text{ unique n-grams } (\hat{x})|}{|\text{total n-grams } (\hat{x})|}\right]$ for a generation $\hat{x}$.

### 6.2.1 RESULTS

Table 2 shows that the SequenceMatch-trained model is able to achieve higher MAUVE score compared to the baselines. It also improves with respect to the n-gram entropy. On the diversity metric, all models are similar, except the contrastive sampling model. The SequenceMatch-trained model is outperformed on the perplexity metric by the BC and MLE-trained methods. This is expected, as the training objective for BC and MLE is exactly the log-perplexity. However, on the MAUVE score, only unlikelihood and SequenceMatch offer a substantial improvement to MLE. Of course, unlikelihood relies on a heuristic to penalize repetitions: a heuristic not appropriate in e.g. arithmetic.

## 7 CONCLUSION

We address the compounding error problem in autoregressive sequence generation by formulating the problem in the imitation learning framework, deriving a general non-adversarial objective for minimizing divergences between occupancy measures induced by a learned model and the data distribution. We develop a novel masking scheme to train a transformer-based autoregressive model with a backspace action with small overhead vs MLE, further reducing compounding error by allowing backtracking. Empirically, the SequenceMatch objective leads to improvements over MLE at text generation and arithmetic. Future work could investigate how qualities of generations change with choice of divergence, or find methods to reduce the overhead of the SequenceMatch objective.

## 8 ACKNOWLEDGEMENTS

This research was supported by funding from the following: Stanford HAI, NSF(#1651565), ARO (W911NF-21-1-0125), ONR (N00014-23-1-2159), and the CZ Biohub. We thank Anuj Nagpal, Div Garg and Andy Shih for valuable discussions and feedback on this research direction.

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

# A ALGORITHM A

---

**Algorithm 2:** Algorithm A: Pseudocode for converting action sequences to masked inputs

---

**Input** : Sequence of action inputs $a_{1:L}$

**output:** Sequence of labels, inputs, masks, position ids

$\qquad y \in |V|^L, x \in |V|^L, m \in \{0,1\}^{L \times L}, p \in [L]^L$

Initialize $y, m, p$ to zero.

Initialize $x$ to $a$.

Initialize $c = 0$                                    `// Copy Pointer`

Initialize $d = 0$                                  `// Deletion Pointer`

**for** $i = 0, \ldots L$ **do**

    $m[i] \leftarrow m[\max(i-1, 0)]$

    **if** $a[i] =$ `<bkspc>` **then**

        $m[i, c] \leftarrow 0$

        $m[i, d] \leftarrow 0$

        $m[i, i] \leftarrow 1$

        $x[i] \leftarrow x[c]$

        $p[i] \leftarrow p[c]$

        $d \leftarrow i$

        $c \leftarrow$ element of last nonzero element in $m[i, 0 : c]$, else 0.

    **end**

    **else**

        $m[i, i] \leftarrow 1$

        $d \leftarrow d + 1$

        $c \leftarrow$ element of first nonzero element in $m[i, c+1 : L]$, else 0.

        $p[i] \leftarrow p[d-1] + 1$

    **end**

    **if** $i = 0$ **then**

        $c \leftarrow 0$                         `// Special cases for initial steps`

        $d \leftarrow 0$

    **end**

    **if** $i = 1$ **then**

        $c \leftarrow 0$

        $d \leftarrow 1$

    **end**

**end**

---

In algorithm A we give a method to convert a sequence of actions into a masked sequence of inputs and corresponding labels, masks and position ids. Although it can also be implemented in a stack-based fashion, we write it as an imperative algorithm so it can be compiled with the `jit` operation in JAX or the `compile` operation in PyTorch. Recall that the idea is to replace a sequence of actions with a sequence of inputs and corresponding labels, masks and position ids. The input sequence at position $t$ should correspond to the state at position $t$ (when the corresponding mask is applied) and the label at position $t + 1$ is the action taken at position $t$. A consequence of this is that the inputs should never include a `<bkspc>` token. The main idea is that if we have a sequence of actions `[..., a, b, <bkspc>, ...]`, the corresponding inputs are `[..., a, b, a, ...]`, while the masks for the second `a` onwards mask out the first `a, b`. However, the possibility of multiple backspaces introduces some complexity.

The approach of the algorithm is to keep a running copy pointer and deletion pointer. The deletion pointer points to the cell of the mask that must be zeroed out for subsequent positions in the sequence, while the copy pointer points to the position that must be copied to the current cell of the input (and also zeroed out in the mask). When a backspace occurs, the deletion pointer is set to the current index, and the copy pointer is sent backwards to the last non-deleted position. When a backspace doesn't occur, the deletion pointer is incremented by 1 and the copy pointer is moved forwards to the first non-deleted position.

# B  Motivating Example Algebra

We consider the case of a length-$n$ Markov chain with an additional node coming from each node. These nodes correspond to the dashed nodes in figure 1. We write the dashed nodes as $x_{\text{term}}$. As in the figure, we have $P_{\text{data}}(x_{\text{term}}) = 0$, $P_{\text{model}}(x_{\text{term}}) = \epsilon$. We wish to compute the divergence between the two distributions.

## B.1  KL-Divergence

We have

$$\mathrm{D}_{\mathrm{KL}}(P\|Q) = \mathbb{E}_{x\sim P}\left[\log P(x) - \log Q(x)\right] = -n\log Q(1-\epsilon).$$

## B.2  Reverse KL-Divergence

We have

$$\mathrm{D}_{\mathrm{KL}}(Q\|P) = \mathbb{E}_{x\sim Q}\left[\log Q(x) - \log P(x)\right] = \infty,$$

since $P(x_{\text{term}} = 0)$ and $Q(x_{\text{term}}) \neq 0$.

## B.3  $\chi^2$-Divergence

We have

$$\mathrm{D}_{\chi^2}(Q, P) = \mathbb{E}_{x\sim Q}\left[\left(\frac{P(x)}{Q(x)} - 1\right)^2\right]$$

# C  Proofs for section 3.2

## C.1  Proof for Saddle Point Theorem

We wish to show that
$$\inf_{\rho_\theta}\sup_r L(\theta, r) = \sup_r \inf_{\rho_\theta} L(\theta, r).$$
Now, the set $\mathcal{D}$ of occupancy measures stemming from conditionals is compact and convex, since it is formed from linear constraints: firstly $\rho \geq 0$, and secondly $\sum_a \rho(s, a) = \gamma \sum_{s'} \rho(s', a) P(s|s', a), \forall s, s'$. Because $\mathcal{D}$ is a closed subset of the infinite-dimensional simplex $\Delta_0^\infty$, which is compact (Rizzolo & Su, 2007), $\mathcal{D}$ is also compact. The set $\mathcal{R}$ is convex, since it consists of all sequences. Since the inner function is convex in $\rho_\theta$ and concave in $r$, we can apply Sion's minimax theorem (Sion, 1958) to swap the inner inf and sup.

## C.2  Proof for Bijection between $r$ and $Q$

Recall that we define the Bellman operator as $\mathcal{B}_r^\theta$, where $\mathcal{B}_r^\theta Q(s, a) = r(s, a) + \gamma\mathbb{E}_{s'\sim\mathcal{P}(s,a)}\left[V^\theta(s')\right]$, for the value function $V^\theta(s) = \mathbb{E}_{a\sim p_\theta(\cdot|s)}[Q(s, a) - \log p_\theta(a|s)]$. The inverse Bellman operator $\mathcal{T}^\theta$ is defined as $(\mathcal{T}^\theta Q)(s, a) = Q(s, a) - \gamma\mathbb{E}_{s'\sim\mathcal{P}(s,a)}\left[V^\theta(s')\right]$.

**Theorem C.1.** *For a fixed policy $\theta$, the inverse soft Bellman operator $\mathcal{T}^\theta$ is bijective, and for any $r \in \mathcal{R}$, $Q = (\mathcal{T}^\theta)^{-1}r$ is the unique fixed point of the Bellman operator $\mathcal{B}_r^\theta$.*

*Proof.* The proof is very similar to the proof of lemma 3.2 in Garg et al. (2021). We construct an infinite matrix $P^\theta \in \mathbb{R}^{(\mathcal{S}\times\mathcal{A})\times(\mathcal{S}\times\mathcal{A})}$, where $(P^\theta f)(s, a) = \mathbb{E}_{s'\sim\mathcal{P}(\cdot|s,a), a'\sim p_\theta(\cdot|s')}[f(s', a')]$. The matrix $P^\theta$ corresponds to the transition matrix for the given MDP and the policy $\theta$. We then have $r = \mathcal{T}^\theta Q$, for any $Q$. Then, $\boldsymbol{r} = \boldsymbol{Q} - \gamma P^\theta(\boldsymbol{Q} - \log \boldsymbol{p_\theta})$. Rearranging, we get $\boldsymbol{Q} = (I - \gamma P^\theta)^{-1}(\boldsymbol{r} - \log \boldsymbol{p_\theta}) + \log \boldsymbol{p_\theta}$. We can do this since $|\gamma P^\theta|_\infty < 1$ if $\gamma < 1$, and so $I - \gamma P^\theta$ is invertible, even in this infinite-dimensional setting. We also see that $\boldsymbol{Q}$ has a unique vector expansion $\boldsymbol{Q} = \boldsymbol{r} + \gamma P^\theta(\boldsymbol{Q} - \log \boldsymbol{p_\theta})$. Since this is the (unique) vector expansion of $\mathcal{B}_r^\theta$, we have $Q = (\mathcal{T}^\theta)r = \mathcal{B}_r^\theta Q$ $\qquad\square$

## C.3 Telescoping Sum Proofs

In this section we prove various theorems related to telescoping sums and value functions. These mostly follow from Kostrikov et al. (2019) and Garg et al. (2021).

**Proposition C.2.** *For a policy $p_\theta$, initial state distribution $\mathcal{P}_0$, value function $V^\theta(s) = \mathbb{E}_{a \sim p_\theta(\cdot|s)} [Q(s,a) - \log p_\theta(a|s)]$, the following identities hold:*

$$\mathbb{E}_{s,a \sim \rho_\theta} \left[ (\mathcal{T}^\theta Q)(s,a) \right] + H[\rho_\theta] = (1-\gamma)\mathbb{E}_{s_0 \sim \mathcal{P}_0} \left[ V^\theta(s_0) \right] \tag{5}$$

$$= \mathbb{E}_{s,s' \sim \rho} \left[ V^\theta(s) - \gamma V^\theta(s') \right], \tag{6}$$

*where $\rho$ is any occupancy measure, and $s, s' \sim \rho$ denotes sampling $s, a$ from $\rho$ and $s'$ from $\mathcal{P}(a,s)$.*

*Proof.* We have

$$\mathbb{E}_{s,a \sim \rho_\theta} \left[ (\mathcal{T}^\theta)Q(s,a) \right] + H[p_\theta] = \mathbb{E}_{s,a \sim \rho_\theta} \left[ Q(a,s) - \gamma \mathbb{E}_{s' \sim \mathcal{P}(s,a)} \left[ V^\theta(s') \right] - \log p_\theta(a|s) \right] \tag{7}$$

$$= \mathbb{E}_{s,s' \sim \rho_\theta} \left[ V^\theta(s) - \gamma V^\theta(s') \right]. \tag{8}$$

By the definition of the occupancy measure and expanding, we have

$$\mathbb{E}_{s,s' \sim \rho_\theta} \left[ V^\theta(s) - \gamma V^\theta(s') \right] = (1-\gamma) \left[ \left[ \mathbb{E}[V^\theta(s_0)] - \gamma \mathbb{E}[V^\theta(s_1)] \right] + \gamma \left[ \mathbb{E}[V^\theta(s_1)] - \gamma \mathbb{E}[V^\theta(s_2)] + \dots \right] \right] \tag{9}$$

$$= (1-\gamma)\mathbb{E}[V^\theta(s_0)]. \tag{10}$$

Because $s_0$ does not depend on $\rho$, we can expand the sum in the opposite direction to show that $\mathbb{E}_{s,a \sim \rho_\theta} \left[ (\mathcal{T}^\theta)Q(s,a) \right] + H[p_\theta] = \mathbb{E}_{s,s' \sim \rho} \left[ V^\theta(s) - \gamma V^\theta(s') \right]$ for any occupancy $\rho$. □

## C.4 Proof of equivalence of solutions of $\mathcal{J}$ and $L$

We now reproduce a proposition from Garg et al. (2021),

**Proposition C.3.** *In the $Q$-policy space, there exists a unique saddle point $(p_\theta^*, Q^*)$, that optimizes $\mathcal{J}$. That is, $Q^* = \arg\max_{Q \in \Omega} \min_{p_\theta} \mathcal{J}(p_\theta, Q)$ and $p_\theta^* = \arg\min_{p_\theta} \max_{Q \in \mathcal{O}} \mathcal{J}(p_\theta, Q)$. Furthermore, $p_\theta^*$ and $r^* = \mathcal{T}^{p_\theta^*} Q^*$ are the solution to the inverse RL objective $L(p_\theta, r)$. This is proposition 3.4 in Garg et al. (2021).*

*Proof.* See Garg et al. (2021) for the complete proof. The proof given applies directly to our case. □

## C.5 Proof for Theorem 3.1

We can now prove our main result

**Proposition C.4.** *With quantities defined in the main text, the following equalities hold for the loss:*

$$\inf_\theta d_\psi(\rho_\theta, \rho_{data}) - H[\rho_\theta] = \sup_r \inf_\theta \mathbb{E}_{s,a \sim \rho_{data}} [r(s,a)] - \mathbb{E}_{s,a \sim \rho_\theta} [r(s,a)] - H[\rho_\theta] - \psi(r),$$

$$= \sup_Q \inf_\theta \mathbb{E}_{s,a \sim \rho_{data}} \left[ (\mathcal{T}^\theta Q)(s,a) \right] - \mathbb{E}_{s,a \sim \rho_\theta} \left[ (\mathcal{T}^\theta Q)(s,a) \right] - H[\rho_\theta] - \psi(\mathcal{T}^\theta Q),$$

$$= \sup_Q \inf_\theta \mathbb{E}_{s,a \sim \rho_{data}} \left[ (\mathcal{T}^\theta Q)(s,a) \right] - (1-\gamma)\mathbb{E}_{s_0 \sim \mathcal{P}_0} \left[ V^\theta(s_0) \right] - \psi(\mathcal{T}^\theta Q),$$

$$= \sup_Q \inf_\theta \mathbb{E}_{s,a \sim \rho_{data}} \left[ \phi(Q(s,a) - \gamma \mathbb{E}_{s' \sim \mathcal{P}(\cdot|s,a)} \left[ V^\theta(s') \right]) \right] - (1-\gamma)\mathbb{E}_{s_0 \sim \mathcal{P}_0} \left[ V^\theta(s_0) \right],$$

$$= \sup_Q \inf_\theta \mathbb{E}_{s,a \sim \rho_{data}} \left[ \phi(Q(s,a) - \gamma \mathbb{E}_{s' \sim \mathcal{P}(\cdot|s,a)} \left[ V^\theta(s') \right]) \right] - \mathbb{E}_{s,s' \sim \rho} \left[ V^\theta(s) - \gamma V^\theta(s') \right],$$

$$= \sup_Q \mathcal{J}(Q) = \sup_Q \mathbb{E}_{s,a \sim \rho_{data}} \left[ \phi(Q(s,a) - \gamma \mathbb{E}_{s' \sim \mathcal{P}(\cdot|s,a)} \left[ V(s') \right]) \right] - \mathbb{E}_{s,s' \sim \rho} \left[ V(s) - \gamma V(s') \right],$$

*Proof.* The first equality is proven in section C.1. The second line follows from sections C.2 and C.4. The first section shows that the objectives $\mathcal{J}(Q, \theta)$ and $L(\theta, r)$ are the same, by the bijective

property of $\mathcal{T}$. The second section proves that the (unique) saddle points of the objectives correspond to the same solutions.

The third line follows from the telescoping sum given in section C.3. The fourth line follows from the substitution of a general $\psi(r)$ with a simpler regularizer $\mathbb{E}_{s,a\sim\rho_{\text{data}}}[g(r(s,a))]$, where $g(r) = r - \phi(r)$ if $r \in \Omega$, and infinity otherwise. This allows us to ground out the divergence minimization directly to concrete divergences such as the KL-divergence, JS-divergence, $\chi^2$-divergence, etc. We discuss this more extensively in section D.1. In the fifth line we expand the telescoping sum in a different way using the result in section C.3. This allows us to incorporate samples from any policy, in order to decrease variance.

In the final line we parameterize the policy from the $Q$-values, setting $\log p_Q(a|s) = Q(s,a) - \log\sum_{a'\in\mathcal{A}}\exp Q(s,a')$. The fact that $\sup_Q\inf_\theta \mathcal{J}(p_\theta, Q) = \sup_Q\mathcal{J}(p_Q, Q)$ follows from the fact that there is a unique saddle point for $\mathcal{J}(p_\theta, Q)$, the fact that $\mathcal{J}(p_Q, Q)$ is concave in $Q$, and that the saddle point for $\mathcal{J}(p_Q, Q)$ has a supremum in $Q$ where $\theta = \theta^*$, with $\log p_\theta^*(a|s) = Q^*(s,a) - \log\sum_{a'\in\mathcal{A}}\exp Q^*(s,a')$ and $Q^*$ the corresponding supremum in $Q$. This allows elimination of $\theta$ from the optimization process entirely, and completes the proof. $\square$

### C.6    PROPERTIES OF THE PLUG-IN ESTIMATOR

The plug-in estimator $\hat{\mathcal{J}}$ is unbiased from the linearity of expectation. Under the assumption that the expected loss is finite, the plug-in estimator is also consistent. This follows from the law of large numbers. We can ensure that the expected loss is finite with the $\chi^2$-divergence by bounding the logits within some large range.

## D    CHOICES OF DIVERGENCE MEASURES

### D.1    $f$-DIVERGENCES

We recall that for any $f$-divergence with $D_f(P, Q) = \mathbb{E}_{x\sim Q}[f(P(x)/Q(x))]$, we have the variational form

$$D_f(P, Q) = \sup_\phi\{\mathbb{E}_{x\sim P}[\phi(x)] - \mathbb{E}_{x\sim Q}[f^*(\phi(x))]\},$$

with the convex conjugate $f^*(y) = \sup_x\{x \cdot y - f(x)\}$) and a discriminator $\phi : \mathcal{X} \to \mathbb{R}$. Optimizing a model against an $f$-divergence other than the KL-divergence typically involves a difficult min-max optimization problem where we simultaneously improve the model and improve the discriminator $\phi$. This is subject to unstable training (Kwon et al., 2021; Jabbar et al., 2020; Tang, 2020; Goodfellow et al., 2014).

In the main paper, we explain that we require a divergence

$$d_\psi(\rho_\theta, \rho_{\text{data}}) = \psi^*(\rho_\theta - \rho_{\text{data}}).$$

With our choice of $\psi$, we get that

$$d_\psi(\rho, \rho_{\text{data}}) = \max_{r\in\mathcal{R}_\psi}\mathbb{E}_{s,a\sim\rho_{\text{data}}}[\phi(r(s,a))] - \mathbb{E}_{s,a\sim\rho_\theta}[r(s,a)]$$

We can readily connect these to $f$-divergences. Recall that the variational formulation of the $f$-divergence is

$$D_f(P, Q) = \sup_g\{\mathbb{E}_{x\sim P}[g(x)] - \mathbb{E}_{x\sim Q}[f^*(g(x))]\}, \tag{11}$$

so we can see that the function $\phi$ we need is simply $-f^*(-x)$.

### D.2    KL-DIVERGENCE

Note that we define our divergence in the reverse fashion to the usual convention, so to obtain the typical forward KL under the expectation of the data, we must use the reverse-KL $f$-divergence,

with $f(x) = x \log x$. This gives $\phi(x) = -e^{-(x+1)}$. However, since we can always shift an f-divergence's $f$ by a constant multiple of $(x - 1)$ without changing the divergence (which should be clear from observing cancellations in the definition of the $f$ divergence), we shift by -1 and (after working through the derivations) have a simpler $\phi(x) = -e^{-x}$.

If we take the objective from equation 4, and observe the limit as $\alpha \to 0,$, we have $\lim_{\alpha \to 0} \mathcal{J}_{\ell_\theta} = D_{\mathrm{KL}}(\rho_{\mathrm{data}} \| \rho_\theta)$. This is because $\lim_{\alpha \to 0} \frac{1}{\alpha} - \exp(-\alpha x) = -1 + x$. Examining the terms in $\hat{\mathcal{J}}(\ell_\theta)$, we can combine the two value sums into a single sum over the data sequence. Then, we can cancel the $\gamma V(s_{i+1})$ terms from the first and second sum. This leaves the term $\sum_i^N \gamma^i(\ell_\theta(a_i|s_i) - V(s_i))$. This is precisely a weighted variant of the typical maximum-likelihood loss.

### D.3 JENSON-SHANNON DIVERGENCE

The Jenson-Shannon divergence has $f(x) = -(x + 1) \log(\frac{x+1}{2}) + x \log x$. This leads to $\phi(x) = \log(2 - e^{-x})$. This is an interesting $\phi$ because it is equal to $-\infty$ for $x < -\log 2$. Since the $x$ in this case is the value of $r$ obtained from the model's logits, it is certainly possible that the value may be less than $-\log 2$. In practice, we could replace $\phi$ with a sharply descending quadratic for all $x$ close to $-\log 2$ and below. This gives a penalizing effect on small $r$, while not causing (too many) numerical issues.

### D.4 $\chi^2$-DIVERGENCE AND $\chi^2$-MIXTURE DIVERGENCE

For the $\chi^2$-divergence, we have $f(x) = ((t - 1)^2)$, leading to $\phi(x) = (x - x^2/4)$.

As described in Al-Hafez et al. (2023), we can add a regularization term by computing $\psi_\rho(r) = \beta c \mathbb{E}_{\rho_{\mathrm{data}}}\left[r(s, a)^2\right] + (1 - \beta) c \mathbb{E}_{\rho_\theta}\left[r(s, a)^2\right]$. In other words, instead of computing the $r^2/4$ term on the expert trajectories only, we also compute this for the policy trajectories as well. We set $c = 0.5$ and $\beta = 0.5$. This results in an even mixture of regularization contributions from the expert and policy. Although this was introduced in Garg et al. (2021) heuristically, it was shown in Al-Hafez et al. (2023) that this has a well-motivated derivation as a result of the divergence between the data occupancy and the mixture between the data occupancy and the policy occupancy:

$$2\chi^2(\rho_{\mathrm{data}} \| \underbrace{\frac{\rho_{\mathrm{data}} + \rho_\theta}{2}}_{\rho_{\mathrm{mix}}}) = \sup_r 2\left(\mathbb{E}_{\rho_{\mathrm{data}}}[r(s,a)] - \mathbb{E}_{\rho_{\mathrm{mix}}}\left[r(s,a) + \frac{r(s,a)^2}{4}\right]\right)$$

$$= \sup_r \mathbb{E}_{\rho_{\mathrm{data}}}[r(s,a)] - \mathbb{E}_{\rho_\theta}[r(s,a)] - c\alpha\mathbb{E}_{\rho_{\mathrm{data}}}\left[r(s,a)^2\right] - c(1 - \alpha)\mathbb{E}_{\rho_\theta}\left[r(s,a)^2\right].$$

In practice, we find this method leads to better quality generations.

### D.5 BACKSPACES ARE THE ONLY EFFICIENT MDPS FOR AUTOREGRESSIVE MODELS

We introduced the backspace as an additional action that we could take now that we do not view autoregressive generation as necessarily modelling a probability distribution. However, we can show that it is a fairly principled choice of action. In fact, with an autoregressive model, the only actions which can be implemented without requiring recomputation of preceding actions are the actions which generate a new token, or do an $n$-step backspace.

To see this, we will first deal with the case of a purely autoregressive model, such as an recurrent neural network (RNN), where processing a sequence $s_i = (x_1, x_2, \ldots, x_i)$ requires sequential processing of every token $x_i$. However, we will assume we keep a buffer of the previous hidden states $(h_1, \ldots h_i)$ so we can revert to these under a backspace. Then, given a current sequence $s$ and an action $a$ which causes a transition to another sequence $s'$, $s$ and $s'$ must differ in at least one token. If $s'$ differs first in a token at location $i \leq |s|$ then we must revert the hidden states to $h_i$ and recompute the next tokens from $i$ to $|s'|$. If $|s'| = i$, then this only requires one pass. This includes the case where no reversion takes place, and $s'$ is $s$ with an additional token appended (i.e. the traditional appending token case). In the case where $s'$ is longer than $s$ by $k$ tokens, we must compute $k$ forward passes. Therefore, the only actions which require one forward pass are those that add a token, or that solely remove a number of tokens.

In the case of a transformer model, the logic is very similar. However, while our analysis is the same as preceding in terms of floating point operations, it is not the same for the latency. As the transformer model can compute a forward pass over multiple input tokens in parallel, an MDP with an action mapping a sequence $s$ to $s'$ could be implemented with relatively low latency, even if $s$ and $s'$ were very different. Whether floating point operations or latency are the main bottleneck will depend on the particular set-up.

## E  ADDITIONAL TRAINING DETAILS

We train each model on four A4000 GPUs with 16GB VRAM each. We keep the batch size at 32 for all models. We use a QLORA $r$ of 64 for all experiments, and an $\alpha$ of 16. Gradient checkpointing and reduced precision was used to reduce memory requirements. For all models, we add an additional row to the unembedding layer of the transformer, corresponding to the logits for the `<bkspc>` action. For the SequenceMatch models, we first train against the BC objective alone for $k$ gradient steps, and then train against a convex combination of the SM loss: $\mathcal{L}_{\text{total}} = \beta \mathcal{L}_{\text{BC}} + (1 - \beta)\mathcal{L}_{\text{SM}}$, where $\beta$ is annealed from 1 to 0.2 linearly over 2,000 gradient steps. For the arithmetic task $k$ is 10,000. For the text generation task $k$ is 1,000. We use a learning rate scheme consisting of a linear warmup from 0 to 2000 steps, followed by cosine decay. For the extra logits offset head, we use a two-layer MLP with Gelu (Hendrycks & Gimpel, 2016) nonlinearities and hidden size equal to 8. The inputs to the layer are the position id and the hidden values at the current position. The output of the layer is added directly to the logits. For SequenceMatch, we keep a replay buffer of past generated sequences. The replay buffer is first-in-last-out, with the oldest sequences being replaced once the size of the buffer is reached. The size of the replay buffer is 1,000. We specify how many times each sequence in the replay buffer will appear in the training data before it is replaced (on average), and from that (and the generation batch size) calculate how frequently a generation step must take place during training. We set the times each sequence will be seen in training to 8. For the text-generation evaluation, we set the prompt length at 256. We then generate sequences of length 256. For the generation, we set the temperature to 1 and the top-p sampling to 1, with no top-k sampling. Since the on-policy regularization term needs the input to be generated from the current policy, we mask the prompt when calculating the regularization for the generated sequences. For the arithmetic task, we mask out the prompt when computing the MLE loss, as this generally leads to more accurate responses (Dettmers et al., 2023). In the arithmetic task, the dataset sequences are constructed as `{question} Solution: {solution}`. For the arithmetic task with 'ground-truth' noise, we add in tokens after the position of the `Solution` token, of digits in the range $0, \ldots, b-1$ for a base-$b$ question. Because the Llama2 tokenizer has separate tokens for combinations of letters such as `a`, `ab`, etc, while having no tokens for combinations of numbers, it would be difficult to construct the ground-truth noise in the bases higher than 10. Therefore, we filter out questions with a base higher than 10. We do not add any noise to the problem statement.

## F  OVERHEAD

In this section we discuss the relative overhead of using the SequenceMatch training objective. We have additional overhead due to the necessity of sampling completions from the model. In addition, the actual loss has some additional complexity compared to the typical MLE loss, as we must extract the terminal values and compute a telescoping sum. Additionally, the on-policy reward regularization term requires computing logits with respect to the generated batch in addition to the data batch. Due to the fact that we use a batch of masks with a complex masking patterns, some GPU kernels that are specialized for causal masking cannot be utilized.

However, we note that it's not necessary to sample at every gradient step, since we can accumulate old trajectories in a replay buffer and re-use them. Furthermore, we can typically sample using a higher batch size than the gradient step batch size, so requiring fewer sampling steps per gradient step. In tables 3 and 4 we show the breakdown of times necessary for a gradient step for each method, in the arithmetic and text modelling cases. We see that the loss computation and gradient step typically takes around two or three times as long for the SequenceMatch objective than the MLE objective, due to the additional computation specified above. The sampling adds additional overhead, depending largely on the length of the sequence that is sampled. We note that the bottleneck from sampling could in principle be completely removed by adding a separate set of GPUs

| Training Procedure | Gradient Step | Sampling Time | Grad Steps per Sample | Total Amortized Time |
|---|---|---|---|---|
| MLE | $1.5 \pm 0.1$ | N/A | N/A | $1.5 \pm 0.1$ |
| SequenceMatch | $5.3 \pm 0.5$ | $50 \pm 10$ | 16 | $8.1 \pm 0.2$ |

Table 3: Execution time of various parts of the training loop for the different models for the 1024 context length, $\chi^2$ objective with model rollouts regularization, with the Llama-2-7b model and 4-bit quantized low-rank training. We show the raw time to sample a batch of trajectories, as well as the time for sampling once amortized due to the fact that we do not sample every training step, and that the training. Because of the unequal memory constraints during quantized low-rank training, we are able to use a large batch size comparatively when generating, allowing us to generate infrequently. Note this is for a batch size per GPU of 1, so each optimization step takes eight times as long for a minibatch size of 32 and 4 GPUs

| Training Procedure | Gradient Step | Sampling Time | Grad Steps per Sample | Total Amortized Time |
|---|---|---|---|---|
| MLE | $1.2 \pm 0.1$ | N/A | N/A | $1.5 \pm 0.1$ |
| SequenceMatch | $2.1 \pm 0.5$ | $2 \pm 0.1$ | 32 | $2.16 \pm 0.1$ |

Table 4: Execution time of various parts of the training loop for the different models for the arithmetic task, $\chi^2$ objective with model rollouts regularization, with the Llama-2-7b model and 4-bit quantized low-rank training. We show the raw time to sample a batch of trajectories, as well as the time for sampling once amortized due to the fact that we do not sample every training step, and that the training. Because of the unequal memory constraints during quantized low-rank training, we are able to use a much larger batch size comparatively when generating, allowing us to generate infrequently. This table is for a batch size per GPU of 8.

which independently generate sequences given the latest parameters and write to a buffer which is polled by the main training thread. Due to time constraints and the additional complexity of synchronizing separate processes, we did not implement this additional distributed training approach.

## G  ADDITIONAL EXPERIMENTS

### G.1  DIFFERENT DIVERGENCES

We briefly experiment with the Jenson-Shannon divergence described in section D.3. As discussed, the main issue is that the function $\phi$ is not defined for inputs less than $-\log 2$, and asymptotically approaches $-\infty$ as the input approaches $-\log 2$. We replace the function $\phi$ with a linear surrogate for $x < -\log 2 + \delta$, with $\delta = 0.01$. The linear surrogate was chosen to be a continuous, differentiable extension of the function $\phi$ from the point $-\log 2 + \delta$.

We kept all the hyperparameters the same as in the experiments with $\chi^2$ divergence. For both the noise settings considered in the main paper, the JS-divergence was only able to achieve 60% accuracy, compared to 85-90% for the $\chi^2$-divergence. There were frequent spikes in the gradient norm compared to the $\chi^2$ divergence.

### G.2  OPUS_BOOKS EN-FR

This translation task was implemented similarly to the arithmetic task in the main paper. The prompt was the English 'question' and the completion being the French 'answer'. We use random noise tokens, with noise level 0.2. The BLEU scores were computed using the `sacrebleu` package. We used a subset of the data consisting of examples where the question was less than 64 tokens long and the answer was less than 64 tokens long. This was still a majority of the examples in the dataset. The BLEU scores are shown in table 5. We see that the SequenceMatch model is able to achieve a small increase in BLEU score compared to the MLE and BC models, although the results have quite high variance. Similarly to the arithmetic experiment, we notice that the backspace is used in generations to correct mistakes, such as the following completion:

English: If he only set two to-day... He would go back to his desk and notice the absence of Meaulnes.

Generation: Si c'était deux qu'il faisait... Il call`<bkspc>` rentrait dans son bureaut`<bkspc>`au, déplorant l'absence de Meaulnes.

We expect that the performance could be improved by using targeted noise tokens, such as common incorrect French translations from a pretrained language model.

| Method | $\eta = 0.02$ | $\eta = 0.2$ |
|--------|---------------|--------------|
| MLE | $31 \pm 2$ | $31 \pm 2$ |
| BC | $32 \pm 2$ | $34 \pm 2$ |
| SM | $37 \pm 4$ | $36 \pm 4$ |

Table 5: BLEU scores obtained by the MLE model, the Behavioral Cloning model, and the SequenceMatch model on the en-fr translation task.

### G.3 ARITHMETIC__MUL

This task was implemented exactly as for the arithmetic addition task, with the same number of training examples. We used only bespoke noise for this experiment. We report the accuracies in table 6. As in the previous experiments, we see an improvement for SequenceMatch, although in this case the behavioral cloning objective does not lead to a significant increase from MLE.

| Method | $\eta = 0.02$ | $\eta = 0.2$ |
|--------|---------------|--------------|
| MLE | $0.49 \pm 0.04$ | $0.49 \pm 0.04$ |
| BC | $0.51 \pm 0.02$ | $0.5 \pm 0.03$ |
| SM | $0.53 \pm 0.03$ | $0.56 \pm 0.04$ |

Table 6: Accuracies obtained by the MLE model, the Behavioral Cloning model, and the Sequence-Match model on the Arithmetic__mul task.

### G.4 NUMBERS__LIST_PRIME_FACTORS

For this task, we initially used 5,000 training data as in the arithmetic tasks. However, we found that all models had very poor performance, with less than 1% accuracy rates. We increased the number of training data to 50,000 and found performance increased. However, the models still had a relatively low rate of accuracy and noisy evaluation, so this experiment may not be very informative. We used bespoke noise at two different levels. The results are shown in table 7.

| Method | $\eta = 0.02$ | $\eta = 0.2$ |
|--------|---------------|--------------|
| MLE | $0.04 \pm 0.03$ | $0.04 \pm 0.03$ |
| BC | $0.07 \pm 0.02$ | $0.06 \pm 0.03$ |
| SM | $0.06 \pm 0.04$ | $0.08 \pm 0.04$ |

Table 7: Accuracies obtained by the MLE model, the Behavioral Cloning model, and the Sequence-Match model on the Numbers__list_prime_factors task.

## H ADDITIONAL RELATED WORK

### H.1 REGULARIZATION TERMS

Due to the disadvantages of the KL-divergence discussed in the first section, several additional training objectives have been proposed which take into account the model's generated sequences.

Particularly popular is the (sequence-level) unlikelihood loss (Welleck et al., 2019). At step $t$, this is given by

$$\mathcal{L}_{\text{ULS}}^t(P_\theta(\cdot|x_{1:t})) = \mathbb{E}_{s_{t+N} \sim P_\theta(\cdot|s_t)} \left[ -\sum_{k=t+1}^{t+N} \sum_{c \in \mathcal{C}^k} P_\theta(c|s_k) \right],$$

where $\mathcal{C}^k$ is a set of problematic outputs, such as repeated $n$-grams. The loss considers an $N$-step sequence generated from the model from $s_t$ and penalizes repetitions. Although this regularization term performs well in practice, it relies on a heuristic that repetitive text is unnatural, which does not necessarily hold universally.

However, the idea of sampling a sequence from the model and including that sequence in the loss is incorporated in SequenceMatch.

## I   EXAMPLES

In this section we give the first few samples from the evaluation set, with no cherry-picking, for the openwebtext task and the SequenceMatch objective. Note that we strip newlines in the examples and generations in order to present them on one page. Furthermore, several unicode characters were generated as random tokens that we cannot represent in the snippet. We label them (U+308A). Cyrillic characters were generated that also cannot be represented, we label them as [ctre] and [nasa] in the samples.

In these examples, we take the prompt sequence and change the last token to a random token drawn from the dataset. We see that the SequenceMatch model is able to utilize the `<bkspc>` token immediately to remove the spurious token. The only case where this does not happen is for the prompt `... A recount is not automatic, operations`, where the random token does make semantic sense, and the model continues by attributing this to a quote from an operations manager.

Table 8: Model Prompts and Completions for the SequenceMatch trained model. We add a random token at the end of the prompt and observe the behavior of the generated sequences.

| Prompt | Completion |
|---|---|
| <bos> Port-au-Prince, Haiti (CNN) -- Earthquake victims, writhing in pain and grasping at life, watched doctors and nurses walk away from a field hospital Friday night after a Belgian medical team evacuated the area, saying it was concerned about security. The decision left CNN Chief Medical Correspondent Sanjay Gupta as the only doctor at the hospital to get the patients through the night. CNN initially reported, based on conversations with some of the doctors, that the United Nations ordered the Belgian First Aid and Support Team to evacuate. However, Belgian Chief Coordinator Geert Gijs, a doctor who was at the hospital with 60 Belgian medical personnel, said it was his decision to pull the team out for the night. Gijs said he requested U.N. security personnel to staff the hospital overnight, but was told that peacekeepers would only be able to evacuate the team. He said it was a "tough decision" but that he accepted the U.N. offer to evacuate after a Canadian medical team, also at the hospital with Canadian securityarono | <bkspc>forces, made a similar decision. He said the military response to the earthquake had been exceptional and he was proud to have worked with them. Gijs had promised the mayor of Port-au-Prince a second field hospital for Friday night. His medical team staffed the hospital from 6 p.m. until about 10 p.m. Friday. But no one came Friday night and there wasnt́ even a security team at the site when the medical team left. Some patients showed signs of serious injuries that were left untreated and an elderly woman is barely conscious. The hospital was in the worst-hit area of the city, and its inhabitants are spending the night in the rubble in the same area. "We were told that they cant́ protect us," Gijs said. "They werent́ even sure they could protect the patients. And some were, you know, just barely alive at this stage and if they walk out and they are attacked, they would never know what happened." Gijs, describing the pullout as "grief," said he had hoped to do something about getting the |
| <bos> Former secretary of state Hillary Clinton meets voters at a campaign rally in St. Louis on Saturday. (Melina Mara/The Washington Post) Democratic front-runner Hillary Clinton was ahead by a slim margin in Missouri on Wednesday, but the race remained in limbo pending word on whether rival Sen. Bernie Sanders of Vermont would seek a recount. The delay postponed a definitive answer to whether Clinton had made a clean sweep of five big primaries on Tuesday night. Even if she does not prevail in Missouri, her other victories push her closer to the Democratic presidential nomination even as the considerably weakened Sanders vowed to press on with his insurgent campaign. Clinton won big in Florida, North Carolina and Ohio, while claiming a narrower victory in Illinois. In Missouri, with 100 percent of precincts reporting, Clinton was ahead 310,602 votes to 309,071. With a difference of less than 1 percent, state officials held off calling the race. A recount is not automatic, operations | manager Michael McLaughlin said. Sanders must first file a request for one with the state elections board. The requests must be filed by noon Friday and paid for by the end of the day on Monday. The Missouri race will linger as a sign of Sanders's political viability even after his latest losses. Sanders did much better than polls had predicted in Missouri, where Clinton maintains close ties to the Missouri Democratic Party establishment and retains the loyalty of much of the state's superdelegate-rich electorate. Clinton's performance Friday in Connecticut, where she won by double digits, may have given her campaign the breathing room it needs to start preparing for the general election. [Clinton cruises to key wins in four large states] Both Clinton and Sanders argued that the latest contests showed that Tuesday night's results merely added to their delegate lead and they didn't provide a game-changing test. The large contests in the South, East and Midwest suggested that Sanders still has a chance to compete nationally, while Clinton can mobilize greater support as she turns to campaigning in more states with |
| <bos> The opinions expressed by columnists are their own and do not represent the views of Townhall.com. You have to give President Barack Obama credit for one thing: consistency. Nothing is ever his fault. Nothing will ever be his fault. Faulting Fox News and the American people, on the other hand, now that's a different story. Do you remember when Obama traipsed around the country and desperately pleaded with Americans to vote for Hillary Clinton because his agenda and his legacy were on the ballot? He made a similar pitch before the shellacking his party took in the 2014 congressional elections. Yet did he acknowledge after this 2014 failing that he had anything to do with it? Does he own up to his leading role in last month's presidential election? Let's rewind the tape further, to Obama's reaction to his party's stunning defeat in the 2010 congressional elections, which was largely about Obamacare. He didn't acknowledge any personal culpability for visiting that monstrosity on the(U+308A) | <bkspc> American people. He didnt́ recognize the failures of his big-government policies. So far so good for Obamaś legacy, his progeny. So far the lessons of this last presidential election have not made their way into the Oval Office. "This election was a turning point, a decisive moment, a determined moment that means we cant́ go back to the policies that got us here. Itś time to turn the page," he said in an interview with Al Arabiyaś Yahya Al-Falluji. In a note-to-self memo, Obama wrote: "Hillary Clinton lost the election because of me." When pressed about how American votersd́esire for policies that are more in line with his big-government presidency, Obama blamed it on people who get their information from the "right-wing media." And if you think this is a sudden realization, a Trump-induced conversion moment, think again. Itś been part and parcel of the leftś worldview ever since polls were first projected on a football field screen more than 1 |
| <bos> BIGBANG is one of those musical entities that transcends language. It's one of those rare groups that both innovates and defines the direction a genre takes. Covering a sound that includes hip hop, R&B and electronic dance, BIGBANG and its solo acts (G-Dragon, T.O.P, Taeyang, Seungri and Daesung) have left a musical imprint that has affected the global music market. In fact, even Diplo, a household name in EDM, worked with G-Dragon and T.O.P for their rap album. So when the band announced its world tour to promote the release of its third full-length studio album MADE after a 3 year hiatus, fans lost their minds { including myself. In fact, tickets for each of BIGBANG's North American legs sold out. As a result, I was lucky enough to witness this larger-than-life Korean pop group perform a couple Saturday nights ago on Oct. 10 at the Prudential Center in Newark, NJ. As I waited in line to enter the venue with my friends, mobs of fans[ctre] | <bkspc> streamed in with BIGBANG snuggie shirts and shirts featuring GD, TOP, Taeyang and Daesung. It was no surprise that the majority of the music being played outside the venue were BIGBANG's hits as well as covers of G-Dragon's solo releases. We spent most of the night standing on the main floor as music permeated the crowds. While other songs were being played, the main focus was on BIGBANG's own music. I found myself moving in time to the dance moves that flowed in a smooth, provocative sexual choreography that found G-Dragon snapping his fingers and sweeping his knees while TOP remained stationary in a regal stance in a white blush jacket. As Daesung appeared on stage, the floor broke into a shriek of chants: \Dae dae! Dae dae!" As the moment passed and Daesung climbed a scaffolding with a blue light backdrop, the crowd screamed again, \Dae dae! Dae dae!" As the chants echoed over and over again, |
| <bos> WHAT?!??! I know. That's what you're saying right now. \WHAT?! DISNEY HAS A DONUT SUNDAE AND I DIDN'T KNOW ABOUT IT?!" How do I know you're saying that? Because that's exactly what I was saying when Tina (<{ amazing DFB photographer who's frigging awesome) spotted this at the Plaza Ice Cream Parlor in the Magic Kingdom this week! But it's OK. It's brand new | not even on the menu yet | so we didn't miss out on too much of the Donut Sundae lifespan. And we're hoping that lifespan is a nice, long one! The Main Street Plaza Ice Cream Parlor can be found at the intersection of Main Street USA and Tomorrowland, just before you get to Cinderella Castle. And the sundae joins a few other must-have treats on the Ice Cream Parlor's menu, including the house-made ice cream sandwich (preferably ordered with a drizzled sauce[nasa] | <bkspc> and fresh topping or two), the house-made vanilla ice cream and the Snickerdoodle Milkshake. The $7.00 Donut Sundae features a house-made vanilla ice cream topped with chocolate covered flakes, chopped bits of fudge, a few walnuts (we think), a generous amount of whipped cream, the choice of mini or jumbo donut, and chocolate or strawberry syrup drizzle. And did I mention CHOCOLATE COVERED FLAKES?!?! Be prepared to shell out between $7 and $8 for this mega Donut Sundae. But you'll probably find yourself licking your fingers of that heavenly chocolate flake as you down the treat, so I'd say it's worth it. At least once. Yep, there it is on the ice cream Parlor menu now. My mouth is watering, just remembering all those amazing chocolate flakes! Do you love chocolate flakes? Who will be the first to give this sundae a try? |