# OpenReview forum: "SequenceMatch: Imitation Learning for Autoregressive Sequence Modelling with Backtracking"
_ICLR.cc/2024/Conference — ICLR 2024 poster_

### Official Review · Reviewer_aeRK · 2023-10-29

**Soundness:** 3 good
**Presentation:** 3 good
**Contribution:** 2 fair
**Rating:** 6
**Confidence:** 3

**Summary:**

This paper introduces SequenceMatch to alleviate the compounding error problem in large language model generation.
First, it introduces a backspace action token to delete the inappropriate token generated in the last step. Second, it introduces a new training objective that minimizes the occupancy divergence between the dataset and the learned model, instead of simply maximizing the likelihood of the ground truth next tokens in the output distribution. Experimental results show that SequenceMatch can improve the accuracy in the arithmetic add-or-sub-in-base sub-task of the match dataset, and improve the mauve score in the openwebtext dataset.

**Strengths:**

- The motivation is convincing
- The method is novel
- The paper writes well

**Weaknesses:**

- The experiment section is not so convincing. The method is evaluated mainly in two settings. In the first setting, the method shows a significant improvement on a subtask of the math dataset. But how about the other sub-tasks? We don’t know if the method can generalize well on other tasks. In the second method, the method has a slightly better mauve score, but the perplexity score drops by a large margin. How can we know that this method indeed improves the model generation?
- The method relies on negative sampling, but there is no reasonable negative sampling strategy provided for open language tasks. This paper provides two negative sampling strategies: random noise and ground-truth noise. As the authors also mentioned in Sec. 6.1, the ground-truth noise needs to be manually constructed for different tasks. The random noise simply samples other tokens in the vocabulary. Both strategies look hard to generalize to open tasks, as it is almost impossible to design ground-truth noise for all the tasks in the world, and the vocabulary for open tasks is too large (the whole token codebook) for the random noise to get meaningful negative samples easily. Thinking about applying SeqenceMatch to the pretraining of LLMs like Llama2, what negative sampling strategy should be used?

**Questions:**

Does the method affect the combinational generalization? Given the famous avocado armchair as an example. Assume we have a training dataset that contains avocados, and different types of armchairs but doesn’t contain avocado armchairs. Does the method discourage the model from combinationally generalizing to avocado armchairs, as it is not included in the training set?

---

> ### Author Response · Authors · 2023-11-17
> **Response to reviewer aeRK**
>
> Thanks for your insightful review. Similarly to reviewer TvsS, you touch on some very important points relating to the tradeoff between different divergences. We answer your specific questions below:
>
> "__In the second method, the method has a slightly better mauve score, but the perplexity score drops by a large margin.__"
> + We agree that the behavior of the perplexity is counter-intuitive. However, in our setting the perplexity does not necessarily correspond to the probability of generating a sequence. We use the term perplexity in the usual way: to denote the exponential of the average negative log likelihood of the next action. Usually the perplexity on the train set directly corresponds to the probability of generating in-distribution sequences. However, in the case with a backspace token, this is not necessarily true, since the probability of generating a sequence must be computed by marginalizing over all the possible sequences of actions (including backtracking via the backspace) that could generate the sequence.
> + As an example, if we consider a very simple language with tokens ‘A’ and ‘B’, where the dataset consists only of sequences ‘AAAAA…’, we can generate an in-distribution sequence by following a policy of sampling A,B uniformly, and then choosing <backspace> if we last sampled a B. We will eventually always generate ‘AAAAA…’, but the perplexity on the dataset is not 1, as it would be for a simple autoregressive model that always generated ‘AAAAA..’. Indeed, a drawback of our method is that we lose the ability to compute the probability of generating a sequence. However, in practice these probabilities do not tend to be very useful for e.g. anomaly detection, so we think it is a good trade-off for the ability to backtrack.
>
> "__There is no reasonable negative sampling strategy provided for open language tasks.__"
> + One possible sampling strategy is the uniform random strategy. As seen in figure 3 and table 2, this has advantages over no added noise. Figure 3 also shows that problem-specific noise can be advantageous. For the problem of open language tasks, uniform noise would be the default, but potentially there could be ways to develop tailored noise, although these are speculative for the time being. For instance, if we assume access to an LLM of similar size, we could use it as an oracle to generate eg incorrect French words in a translation task. We could also use crowd workers to generate text and record the complete generative process, including any backspaces used by the workers.
>
> "__Thinking about applying SeqenceMatch to the pretraining of LLMs like Llama2, what negative sampling strategy should be used?__"
> + As mentioned above, the random uniform sampling is a baseline approach which gives some advantage. Other approaches could be investigated, including using other models to generate plausible (but incorrect) continuation tokens.
>
> "__Does the method discourage the model from combinationally generalizing to avocado armchairs, as it is not included in the training set?__"
> + This is a great question. Unfortunately, we don’t have a way to answer it theoretically; at least, we’re unaware of a mathematical formalism of the combinatorial generalization phenomenon. Intuitively, we think it’s likely that the loss encourages more mode-concentrating behavior in exchange for increased probability of staying in-distribution, as discussed in the response to reviewer TvsS. This would lead to decreased levels of ‘combinatorial generalization’. Of course, in the setting of language models, we could argue that hallucinations are an example of incorrect generalization, where a model is happy to continue generating details of events that did not happen. Perhaps in this setting, or settings such as question answering, it would be better if the level of generalization was lower.
>
> Please let us know if there are any remaining questions; we would be happy to discuss them!

---

> > ### Comment · Reviewer_aeRK · 2023-11-22
> > **Response to author comments**
> >
> > Thank you for answering my questions! The explanation of why real perplexity is hard to compute in the proposed method makes sense. However, since perplexity is one of the most important metrics in LLMs, is it possible to construct some approximation (e.g., by marginalizing over cases where there are zero, one or two <backspace> action in the response) and see if the perplexity score can approach the baseline?

---

> > > ### Author Response · Authors · 2023-11-23
> > > **Response regarding perplexity**
> > >
> > > We’re glad to hear that our answers were useful! Your suggestion is very interesting. Concretely, this means that if we want to compute $P(x_1, x_2, x_3)$, we can approximate this by considering terms with zero, one or two backspaces. If we consider a term with one backspace, we would have to compute e.g. the term $P(x_3 | x_1, x_2)$ by calculating $P(x_3 | x_1, x_2)( 1 + \sum_{x_3' \in V}P(x_3’|x_2,x_1)P(<\text{bkspc}> | x_3', x_2, x_1))$, i.e. summing over the cases where there is a backspace as the next token.
> > >
> > > There are a few challenges with this approach. Firstly, the exact sum is not really feasible to compute since it would need $|V|$ forward passes through the model to compute $P(<\text{bkspc}>|x_3', x_2,x_1)$, where $|V|$ is the size of the vocabulary. For Llama-2 $|V|$ is around 30,000, so would require about the same number of forward passes to marginalize this one conditional as to fine-tune for the whole arithmetic task.
> > >
> > > Now, it would be possible to approximate this marginalization by sampling, approximating $\sum_{x_3' \in V}P(x_3’|x_2,x_1)P(<\text{bkspc}> | x_3', x_2, x_1))$ with a sampled estimator of  $\mathbb{E}_{x_3' \sim P(.|x_2,x_1)}\left[P(<\text{bkspc}> | x_3', x_2, x_1))\right]$. This would still require on the order of $kL$ forward passes for an input of length $L$ and sampling $k$ alternatives for each conditional, compared to one forward pass for the typical perplexity evaluation.
> > >
> > > A (maybe more theoretical than practical) challenge with the one-backspace approximation is that it’s biased, in the sense that it will under-estimate the probability of a sequence. Consider the ‘AAAA’ example above, with the policy of sampling A w.p. 50% and B w.p. 50%, but always sampling <bkspc> if the last character was B.
> > > Looking at the sub-sequence ‘AA’, we compute P(‘AA’) as P(A|A)(1 + P(<bkspc>|BA)P(B|A)) = 0.5 * (1 + 0.5) = 0.75, with the 1-backspace approximation. The full marginalized probability is found by summing the probabilities of sequences of actions ‘AA’, ‘AB<bkspc>A’, ‘AB<bkspc>B<bkspc>A’, which are 0.5, 0.25, 0.125, etc. In other words, any truncation after a finite number of backspaces will under-value the probability of generating the sequence. Following this logic, the quantity we report under ‘perplexity’ for the sequencematch model is necessarily an over-estimate of the model’s ‘true perplexity’
> > >
> > > Although we’re not able to implement and run this approximated marginalized perplexity due to lack of time before the end of the discussion period, we will aim to do so in the next version of the paper.
> > >
> > > We do want to point out, however, that in other areas of generative models such as image generation, it is common to use non-likelihood-based methods of quality evaluation such as the FID score. Similarly, it has also been shown that models such as GANs can generate very good quality samples with poor likelihoods [1]. Given our model uses alternative divergences which emphasise more mode-covering behavior (as discussed in the discussion with reviewer TvsS), it’s not particularly surprising that for a fixed model complexity, focussing on more mode-collapsing divergences comes at the expense of likelihood. However, as we try to emphasize throughout the paper, from a practical point of view, it’s often better in downstream tasks to have a model which generates coherent text with less diversity than a model which is diverse but cannot generate long sequences.
> > >
> > > 1. Flow-GAN: Combining Maximum Likelihood and Adversarial Learning in
> > > Generative Models, Aditya Grover, Manik Dhar, Stefano Ermon, AAAI 2018

---

### Official Review · Reviewer_roYx · 2023-10-31

**Soundness:** 3 good
**Presentation:** 4 excellent
**Contribution:** 3 good
**Rating:** 6
**Confidence:** 3

**Summary:**

The paper addresses the problem of learning autoregressive models for (discrete) sequence modeling. It identifies two drawbacks of traditional maximum likelihood approaches: Firstly, such techniques, like behavioral cloning, allocate weights to sequences based on their prevalence in the training set. They do not consider the problem of guiding the autoregressive model when it starts to be out-of-distribution. To tackle this issue, the authors draw inspiration from Garg et al. 2021 and put forth a non-adversarial objective aimed at minimizing multiple divergences. A distinguishing feature from Garg et al. 2021 is their extension of the methodology to infinite-dimensional state spaces. The second drawback is that classical auto-regressive methods can go OOD by predicting bad tokens. To address this, the authors introduce an innovative approach of incorporating a "backspace" token, which has the capability to erase the preceding token. To facilitate the model in learning when to generate this backspace token, the authors suggest the introduction of noise into the training sequences. They then enhance these sequences with the backspace token, employing the modified sequences as training data. Additionally, they present a systematic method to process the backspace tokens in transformers using masking. This ensures that the renowned high-speed learning capabilities of transformer models on GPUs remain uncompromised.

They compare their approach with classical approaches on an arithmetic task, and on text generation, with positive results showing the effectiveness of the two components.

**Strengths:**

The paper is very well written and addresses a significant and relevant issue. It offers a solution designed to minimize various divergences associated with sequence generation. Notably, the authors introduce the innovative concept of a "backspace" token, alongside presenting a proficient method to seamlessly incorporate this token into the training set. Furthermore, the experimental outcomes presented are both compelling and convincing. The contributions may be of the interest of a large audience.

**Weaknesses:**

The paper proposes two distinct contributions. Firstly, it extends the work of Garg et al. 2021 to handle an infinite-dimensional state space. However, based on my understanding, this extension doesn't seem particularly pertinent to the problems explored in the experimental section. While I'm somewhat familiar with Garg et al.'s work (having read it for this review), the nuances differentiating the two papers aren't immediately clear to me. I'd encourage the authors to better identify these differences more explicitly. While Garg et al. 2021 delves into reinforcement learning, the authors of the current paper narrow their focus to sequence generation, which can be seen as a simpler challenge than the broader realm of RL. The originality of this contribution remains somewhat ambiguous to me.

The paper's second contribution (introducing backspace) is more algorithmic and makes sense, especially given the efficient implementation strategy the authors have introduced. However, this contribution stands independent of the first, resulting in a paper that seems to amalgamate two separate ideas. While the combined nature of these contributions makes the proposed model efficient, it also emphasizes that individually these contributions are not so strong.

(Small remark: bibliography is in the supplementary material and has to be moved back to the main paper)

**Questions:**

* Please better identify what is the contribution w.r.t Garg et al. 2021

**Details Of Ethics Concerns:**

No concerns

---

> ### Author Response · Authors · 2023-11-17
> **Response to reviewer roYx**
>
> Thanks for your thoughtful and comprehensive review. Your main question was about the relation of the two contributions, and the novelty of the approach compared to Garg [2021]. We hope to have answered this question in the global response.
>
> __Please better identify what is the contribution w.r.t Garg et al. 2021__
> + See global response
>
> Please let us know if there are any remaining questions or aspects of the questions we have not addressed, and we would be very happy to discuss them!

---

> > ### Author Response · Authors · 2023-11-23
> >
> > We kindly draw your attention to our rebuttal, and would greatly appreciate your response before the end of the discussion period.
> > We are happy to address any additional questions, and once again thank you for your time and consideration.

---

### Official Review · Reviewer_n9EH · 2023-11-09

**Soundness:** 2 fair
**Presentation:** 3 good
**Contribution:** 3 good
**Rating:** 6
**Confidence:** 2

**Summary:**

This paper formulates autoregressive sequence generation as an imitation learning (IL) problem. To tackle the compounding error issue brought by the original MLE objective, it minimizes the occupancy divergence between the sampled sequences from the model and the data distributions. The paper further proposes a masking scheme that allows generation with a backspace action.

**Strengths:**

1. This paper offers a novel perspective for sequence modeling by formulating it as an IL problem. It tackles the error compounding issue, a significant drawback of sequence modeling, by adopting a divergence minimization approach.
2. The backspace action is novel for sequence modeling. Allowing backtracking during decoding may greatly benefit tasks in reasoning and planning.
3. The experiments demonstrate using backspace improves the accuracy and detects OOD cases.

**Weaknesses:**

1. The authors only present results on two tasks. It would be more convincing to add more results on, e.g., summarization, translation, and more complicated reasoning tasks.
2. The method induces nonnegligible training overhead due to sampling trajectories and using a complex loss objective. Also, using backspace introduces some inference overhead -- is the case of rolling back a long segment very unlikely?
3. The proposed formulation in Sec 3 seems to be an application of Garg et al. (2021) to the case of sequence modeling. May authors elaborate the technical novelties w.r.t. Garg et al. (2021)?

**Questions:**

See weakness.

**Details Of Ethics Concerns:**

No ethic concerns.

---

> ### Author Response · Authors · 2023-11-17
> **Response to Reviewer n9EH**
>
> Thanks for your helpful review. We answer your questions below.
>
> "__It would be more convincing to add more results on, e.g., summarization, translation, and more complicated reasoning tasks.__"
> + See global response
>
> "__The method induces nonnegligible training overhead due to sampling trajectories and using a complex loss objective__"
> + We give some numerical details on the training overhead in the appendix, section F. The overhead of the loss itself is mainly due to the loss of specialized cuda kernels for causally-masked attention. The overhead of the sampling is fundamentally unavoidable, however. It can be amortized by keeping a replay buffer of previously-generated samples and using those stale replays during training. Since the loss approach is off-policy, using stale replays is valid. If this approach is deployed at a large scale during pre-training, we could also parallelize the generation during training--one machine or set of GPUs could be used to continuously generate sequences, and the parameters for the generation machine updated periodically.
>
> "__Also, using backspace introduces some inference overhead -- is the case of rolling back a long segment very unlikely?__"
> + Yes, it’s unlikely. In fact, even under high levels of noise, the generated sequences do not contain many backspaces. With 20% of tokens subject to additional noise, the rate of backspaces in the arithmetic task was around 1.5% of all actions for the SequenceMatch-trained model. It was rare to have more than one backspace in a row. On the other hand, we found that of the cases where backspace is used, it is around 90% accurate--it only removes a correct token 10% of the time.
>
> "__May authors elaborate the technical novelties w.r.t. Garg et al. (2021)?__"
> + See global response
>
> If there are any remaining uncertainties, we are happy to answer them during the discussion phase.

---

> > ### Author Response · Authors · 2023-11-23
> >
> > We point your attention to our rebuttal, and would greatly appreciate your response before the end of the discussion period.
> > We are happy to address any additional questions, and once again thank you for your time and consideration.

---

> > > ### Comment · Reviewer_n9EH · 2023-11-23
> > >
> > > Thank you for providing detailed comparison with existing works and adding new experimental results. I have no additional questions.

---

### Official Review · Reviewer_TvsS · 2023-11-10

**Soundness:** 3 good
**Presentation:** 3 good
**Contribution:** 2 fair
**Rating:** 6
**Confidence:** 2

**Summary:**

This paper introduces SequenceMatch - a novel minimization objective for training autoregressive models.  The proposed approach trains using the chi^2 divergence rather than the standard KL maximum likelihood (ML) objective.  An estimator for the loss is defined and a training algorithm is given.  A backspace token is added additionally to account for the model generating more OOD sequences given the changed objective.Results are provided for fine-tuning on a math dataset and OpenWebText generation

**Strengths:**

* The proposed work tackles an important problem in the generation on OOD sequences and proposes a new approach which the authors derive and give a practical estimator for.  The ideas were studied in prior VI and generative model literature but appear to be new for language models - the application in this work.
* It is appreciated that the authors have suggested a novel theoretical approach demonstrating that one does not need to do any variational bound, nor adversarial training to compute the loss.  This makes the proposed approach easier to implement and use.  The authors also derived an empirical estimator for practical purpose.  The idea of using the backspace to handle the generation of more OOD sequences is also good, which although the proposed approach introduces a new problem of more diverse generations, the additional backspace helps address this.
* Overall the paper is well-written and motivates itself well from prior literature in imitation learning.

**Weaknesses:**

* The authors propose using the chi-sq divergence but do not motivate using the chi-sq over other f-divergences only citing its use in prior works.  A more natural divergence based on f-divergences and shortcomings of KL is the Jensen-Shannon (JSD), which also has the benefit of being symmetric and adds the reverse KL.  Why have the authors not compared with other divergences or why did these divergences not perform well?
* The proposed work provides an estimator but does not show any properties of the new estimator (consistency, unbiasedness, etc.).  Historically, the MLE has been nice properties, but it is unclear whether Eq (4) has any nice properties with relation to the true divergence.
* Performance of the proposed approach does not perform much better than the MLE except for MAUVE, when the data does not appear to be OOD (OpenWebText).  Additional experiments on OOD datasets (would artifical noise)   would supplement the results and show better performance potentially.
* The SequenceMatch approach is relatively slow taking 1.5-8x the training time.  Due to the number of gradient updates, it may also be prohibitively expensive for full-finetuning.  However, this is not discussed in the paper.

**Questions:**

* Can the authors clarify the Figure 1 example? My intuition is that the MLE should come from KL(p_{data} | p_{model}), which is defined in the example whereas the reverse KL(p_{model} | p_{data}) is not.  Isn't this suggesting the reverse KL is desired?
* There is an implicit assumption in this work that the KL divergence is not a good metric because what we desire is for the model not to generate when the true data distribution has probability 0.  The work does not discuss the opposite, however where the data distribution has non-zero mass, KL minimization will encourage the model to always try to match the distribution - an explicit advantage of KL.  Without minimizing KL my concern is that we do not cover the full data distribution.  Explicitly, to address the main metric of interest in 2.1.1, implicitly the reverse KL would be a good choice to optimize.  This can be a problem especially for a large model or small data.
* Given that the new metric may not cover the full distribution as well as KL minimization evaluation on retrieval or Q/A tasks where information is needed - will further improve the paper.

---

> ### Author Response · Authors · 2023-11-17
> **Response to Reviewer TvsS**
>
> Thank you for your thoughtful and detailed review. You raise several excellent points which we answer here.
>
> "__Why have the authors not compared with other divergences or why did these divergences not perform well?__"
>
> + On prototyping iterations of the project, we tried different divergences and observed that they did not work as well as the $\chi^2$-divergence. In particular, we observed that the Q-values (logits) would become very large and training would be unstable for divergences other than $\chi^2$. Previous work showed that the $\chi^2$ divergence had the strongest performance in high-dimensional tasks (e.g. appendix B.2. in [Garg 2021]). Furthermore, other work such as [Al-Hafez et al 2023] has provided theoretical justification for this improved performance, namely the fact that the optimal Q-values are bounded.
> + From a practical point of view, the reverse KL divergence and JS divergence pose algorithmic challenges, since the function $\phi(x)$ in equation 4 is $1 + \log x$ and $\log (2 - \exp(-x))$, which are only defined for $x > 0$ and $x > -\log 2$ respectively. This would require constraining $\ell(s,a) - \gamma V(s’)$ for all states, actions, next states $s,a,s’$. It’s not obvious how to algorithmically enforce such a constraint given that we are parameterizing $\ell$. Another option is to form a relaxation for $\phi$ with e.g. a very large negative value where $x$ is out of the domain of $\phi$, but this makes optimization very unstable.
> We are attempting ablations with different divergences, using the relaxation-based method, which we will hopefully have available by the end of the author-reviewer discussion period.
>
> "__Does not show any properties of the new estimator (consistency, unbiasedness, etc.)__"
>
> + Thank you for pointing this out. The estimator in equation 4 is a plug-in estimator for the population objective stated in the previous section. As such, under modest conditions it is unbiased and consistent, in the sense that the expected value of ${\hat J}(\ell_{\theta})$ is $J(\ell_{\theta})$ and ${\hat J}(\ell_{\theta})$ converges in probability to $J(\ell_\theta)$ as we increase the number of sequences used in the estimator. For simplicity in equation 4 we present the estimator using a single sequence. In practice we always average the loss over several sequences, with a batch size of 64 typical. Using conditions such as boundedness of the logits and $\phi = x^2 - x^2/4$ for the $\chi^2$-divergence, proof of unbiasedness follows from linearity of expectation, and consistency from the central limit theorem. We will include these proofs in the next version of the appendix.
>
> [Continued in next reply]

---

> > ### Author Response · Authors · 2023-11-17
> > **Response to Reviewer TvsS - Part 2**
> >
> > [Continued from previous reply]
> >
> > "__Can the authors clarify the Figure 1 example?__"
> >
> > + We didn’t explicitly mention this in the paper, but it’s worth keeping in mind that for any divergence $D$, the divergence $D(P,Q)$ is zero if and only if $P = Q$, so in the limit with infinite samples and a universal function approximator, there’s no difference in the solution to the divergence-matching problem for different divergences. As such, the goal of figure 1 is to show that different divergences will penalize different approximately-correct sequence models to different degrees. In this case, the forward KL is approximately proportional to $n \epsilon$, the reverse KL is always $\infty$, and the $\chi^2$-divergence is proportional to the probability of generating an OOD sequence, squared.
> >
> > "__Isn't this suggesting the reverse KL is desired?__"
> > + Yes in a certain sense the reverse KL is the most desirable, since in this example it is always infinite if the model ever generates any sequences which are ‘OOD’, where the expert has zero density. However, in practice this would be a very difficult divergence to optimize with gradient descent over finite samples, as the Lipschitz constant of the loss is infinite. As mentioned above, going through the derivations in section D for the reverse KL, the function $\phi$ in equation 4 is $ 1 + \log x$ for the reverse KL, so if $\ell(s,a) - \gamma V(s’)$ is arbitrarily close to zero, the loss is unbounded.
> >
> > "__Without minimizing KL my concern is that we do not cover the full data distribution.__"
> > + When we can’t match a distribution exactly, any choice of loss is implicitly specifying a preference of some (approximate) solutions over others. For a single conditional, the forward KL will certainly have more mode-covering behavior than the $\chi^2$ or reverse KL. However, as we discuss in section 2.1.1., the compounding error problem can have unwelcome interactions with the mode-covering behavior.
> > + As a simple example, consider a generative process for a sequence $x_1, x_2, \ldots, x_N$, where each $x_i \in \mathbb{R}$. The ground-truth conditional distribution $P(x_{i+1}|x_i)$ is a mixture of Gaussians with two modes. However, let’s say our parameterized conditional model is a unimodal Gaussian. Then, the forward KL will generically have mode-covering behavior with some density outside the bulk of the expert density. The reverse KL or $chi^2$ will generally have more mode-covering behavior. The problem arises when we use the forward-KL-based model to generate sequences, since the mode-covering model will sample points $x_1$ that are out-of-distribution and have been seen very infrequently during training. Meanwhile, the reverse KL trained model (as an extreme example) would have all samples in-distribution, although they would all be in the same mode (so lack diversity).
> > + Certainly it would be better if we could cover the full distribution, but given the choice between covering the distribution to some limited extent, and getting coherent long sequences, many would prefer the coherent sequences, especially for a task such as question-answering, where receiving a coherent but non-diverse response is generally preferred to a set of diverse but incoherent results.
> >
> > "__evaluation on retrieval or Q/A tasks where information is needed - will further improve the paper.__"
> > + Please see the global response
> >
> > Thanks again for the excellent questions; if you have any additional clarifications or points you would like to raise, we’re more than happy to discuss!

---

> > > ### Comment · Reviewer_TvsS · 2023-11-22
> > > **Response to author comments**
> > >
> > > Thank you for addressing the questions raised in the review.
> > >
> > > I appreciate the clarification on the properties of the estimator in Eq.4 and note that the authors have given justification for why the chosen divergence is used over existing divergences both for the imitation learning setup and properties of the divergences. For this reason I am willing to increase my score to above the acceptance threshold.
> > >
> > > Still, I believe the paper is lacking comparison with other divergences - it can't be said empirically that penalization from chi^2 is better. Further, there are claims the authors have highlighted - "especially for a task such as question-answering, where receiving a coherent but non-diverse response is generally preferred to a set of diverse but incoherent results."  - that should be tested to demonstrate that training with the new divergence leads to performance increase on benchmark downstream tasks.

---

> > > > ### Author Response · Authors · 2023-11-23
> > > > **Response to reviewer**
> > > >
> > > > Thank you for taking our response into account, and we’re glad we could address your questions. We agree that evaluation on downstream tasks such as question-answering, with different divergences, are important.
> > > >
> > > > As a preliminary analysis, we ran an experiment on using the arithmetic task using the JS divergence with a relaxation, such that $\phi(x)$ was replaced with a linear surrogate for $x < \log 2 + \delta$, for $\delta = 0.01$. With all other hyperparameters as specified in the main paper, we were only able to obtain an accuracy of around 60% over two random seeds when the noise fraction was 0.2, and the noise used was bespoke noise. This compares to about 90% accuracy for the same settings with the $\chi^2$-divergence. As expected, the training was very unstable, often crashing due to NaNs in the loss or gradient. We will add this analysis to a section in the appendix in the next version of the paper, once we have had time to check more noise settings and different hyperparameters.
> > > >
> > > > We completely agree that the JS divergence or other divergences are reasonable to investigate; however in our particular setup with the $\phi$ function, training against the JS is numerically unstable. We hope that future work could develop different parameterizations or methods to allow training against the JS divergence.

---

### Author Response · Authors · 2023-11-17
**General Response to Reviewers**

# General Comments

We thank all the authors for their thoughtful and insightful reviews. We’re happy to see that reviewers TvsS, roYx and roYx found our work well-written, and all reviewers thought our approach was novel.

In this global response we address two points that several reviewers raised. In the individual reviewer responses we will address points raised by individual reviewers. If there are any further questions or clarifications, we are very happy to discuss them during the remainder of the discussion period.

## Novelty with respect to [Garg 2021]
Both reviewers n9EH and roYx asked us to clarify the extent of our contributions compared to Garg (2021), which introduced the IQ-Learn algorithm. As we point out in the related work section, ‘We can view our approach as a specialization of the IQ-Learn algorithm in Garg(2021) to autoregressive sequence models.’ Indeed, the proof of proposition 4 follows very similarly to the proof in Garg (2021). We view our contribution as having a few different parts:

### Contribution by viewing language generation as an RL task and incorporation of backspace
Viewing sequence generation as an imitation learning/RL task is our first conceptual contribution. Previous work implicitly takes the point of view that minimizing the KL-divergence between the model and data generative model is the correct objective. In this previous work, there is an assumption that minimizing the KL is the best objective if we wish to generate good sequences, even though this is not necessarily true for an arbitrary definition of ‘good’.
Viewing the sequence generation problem as an imitation learning problem is an alternative point of view, where minimizing divergences between generated sequences and expert sequences is the most important concept. We think this ’generation-first’ approach makes a lot of conceptual sense.

Secondly, we contribute the exact formulation of the Markov decision process required to turn sequence generation into an imitation learning problem. It’s straightforward to see the states as partial sequences and actions as appending tokens. However, the inclusion of the backspace as a possible action is not obvious--we could create an MDP without a backspace action. In fact, although we do not emphasise this in the main paper, the theoretical contribution is made much easier with the backspace action. Without the backspace action, the MDP is not ergodic, and so typical mixing results do not hold. In early experiments, we found that the SequenceMatch loss didn’t lead to much improved performance without the backspace token, as it’s not possible to ‘correct’ OOD behavior easily.

The backspace action was added to improve correction of OOD behavior. However, we realized that it is a special and unique action in the context of autoregressive sequence models. In the next version, we will include a brief proof that our choice of dynamics is comprehensive, in the sense that the only MDPs which can be constructed for sequence modelling and can be evaluated in an autoregressive model without recomputation are the append-token, end-of-sentence, and backspace-n-times tokens.

### Contribution in combining backspace and IQ-Learn
Another contribution is in utilizing SequenceMatch to train with the backspace. As described in the main text, it is possible to train with a backspace token in a purely supervised fashion with the behavioral cloning approach. However, this approach does not utilize the dynamics information -- i.e. the knowledge that taking the backspace action in state $s_t$ will lead to the state $s_{t-1}$. We could think up heuristic rules such as ‘if the backspace is taken in state a, we should reduce the probability of going to state a in the future’. The SequenceMatch loss will (in principle) automatically take care of that with its loss, which depends on the current state, action, and next state. Figure 3 in the paper shows that backspace trained with SequenceMatch improves over the performance of backspace trained with behavioral cloning.

### Contribution of Masking Approach
Finally, a main contribution is the masking approach, which allows training of an MDP trajectory rollout with effectively the same complexity as evaluating a supervised loss. This allows us to incorporate the backspace with little overhead (in the loss).

## New Experiments
Several reviewers (TvsS, n9EH) asked for some additional experiments. We are currently carrying out additional experiments: the English-French translation task from the `opus_books` dataset, and the `arithmetic__mul` and `numbers__list_prime_factors` additional tasks from the `math_dataset` dataset. We expect to have these results ready by the end of the reviewer-author discussion period. For each dataset, we are running Behavioral Cloning, SequenceMatch, and MLE with added noise (bespoke and non-bespoke noise where relevant)

---

> ### Author Response · Authors · 2023-11-23
> **Additional Updates**
>
> The additional experiments are as follows, and will be included in the paper once we have more time to run additional hyperparameter sweeps and random seeds. Unfortunately we didn’t have time to do the non-bespoke noise experiments for the arithmetic tasks, although these will be included in the next version of the paper.
>
> ### Opus_books en-fr
> We implemented this translation task similarly to the arithmetic task in the main paper, with the prompt being the `en` ‘question’ and the completion being the `fr` ‘answer’. We use random noise tokens, with noise level 0.2. The BLEU scores were computed using the `sacrebleu` package. In the interests of getting the results before the deadline with our limited resources, we used a subset of the data consisting of examples where the en prompt was less than 64 tokens long and the fr completion was less than 64 tokens long. This was still a majority of the examples in the dataset.
> The BLEU scores were as follows:
>
> MLE: $31 \pm 2$
>
> BC: $34 \pm 2$
>
> SM: $36 \pm 4$
>
> We observe that the bkspc is used in generations to correct mistakes, such as in this completion:
>
> __en__: If he only set two to-day . . . He would go back to his desk and notice the absence of Meaulnes.
>
> __generation__: Si c’était deux qu’il faisait… Il call<bkspc> rentrait dans son bureaut<bkspc>au, déplorant l’absence de Meaulnes.
>
> ### Arithmetic__mul
> This task was implemented exactly as for the arithmetic addition task. For bespoke noise, noise level 0.2, over two random seeds, the accuracies were:
>
> MLE: $0.49 \pm 0.04$
>
> BC: $0.5 \pm 0.03$
>
> SM: $0.56 \pm 0.04$
>
> As before, we see a significant improvement for SequenceMatch, although in this case the improvement for behavioral cloning from MLE is not very large.
>
> ### Numbers__list_prime_factors
>
> For this task, we initially used 5,000 training data as in the arithmetic tasks. However, we found that all models had very poor performance, with less than 1% accuracy rates. We increased the number of training data to 50,000 and found performance increased. However, the models still had a relatively low rate of accuracy, and so this experiment may not tell us very much.
> For bespoke noise, noise level 0.2, the accuracies were:
>
> MLE: $0.04 \pm 0.03$
>
> BC: $0.06 \pm 0.03$
>
> SM: $0.08 \pm 0.04$
>
> ### Additional changes
> We updated the paper to include the bibliography. As pointed out by reviewer roYx, we had accidentally included this with the supplementary material in the original submission.

---

### Meta-Review · Area_Chair_ekF2 · 2023-12-05

**Metareview:**

This paper introduces SequenceMatch to address autoregressive model training by minimizing divergence using a chi^2 metric instead of standard KL maximum likelihood. The addition of a backspace token for handling out-of-distribution sequences is an appreciated innovation. Reviewers generally acknowledge the paper's importance in addressing error compounding issues, offering a unique perspective on sequence modeling. However, concerns exist regarding limited experimental evaluations, especially in different task domains beyond math datasets and OpenWebText. Questions remain about the choice of divergence metric and its comparison with other metrics like Jensen-Shannon. Some reviewers raise concerns about the lack of in-depth analysis of the estimator's properties and the overhead introduced by the proposed approach in terms of training and inference times. Clarifications were provided in the rebuttal, addressing some concerns, but comparisons with other divergences and performance on various tasks, including larger language models, remain areas for improvement. The efficient implementation strategy for the backspace token is acknowledged positively. Despite improvements after rebuttal, further experimentation and exploration of divergences and broader task domains are suggested to strengthen the paper's claims and contributions.

**Justification For Why Not Higher Score:**

The score wasn't elevated due to several key reasons: limited empirical evaluations beyond specific datasets, lack of thorough comparison with alternative divergence metrics, concerns about training and inference overhead, and the perceived separation between the paper's contributions. Although clarifications were provided in the rebuttal, addressing some concerns, further empirical evidence and cohesive integration of contributions are needed to justify a higher score.

**Justification For Why Not Lower Score:**

The paper addresses an important problem in autoregressive model training, offers a novel perspective on sequence modeling, and introduces innovative concepts like the backspace token. Despite some limitations in empirical evaluations and integration of contributions, the paper's well-written presentation and potential impact warrant a score that acknowledges its strengths while highlighting areas for improvement.

---

### Decision · Program_Chairs · 2024-01-16

Accept (poster)